# MULTI-MODALITY IMAGE FUSION UNDER ADVERSE WEATHER: MASK-GUIDED FEATURE RESTORATION AND INTERACTION

## ABSTRACT

Multi-modality image fusion (MMIF) enhances scene representation by exploiting complementary cues from different modalities. Adverse weather, however, causes significant image degradation, disrupting feature representation and requiring simultaneous feature restoration and cross-modal complementarity. Existing methods often struggle with effective representation learning under such conditions, limiting their practical performance. To address these challenges, we propose a mask-guided MMIF method that integrates feature restoration and interaction. We first introduce "Pseudo Ground Truth" to simplify training, promoting faster and more effective feature learning. Then, we design a mask generation mechanism based on the mapping relationship between the fused result and the source images, quantifying the relative contribution of each modality during the fusion process. By incorporating the proposed mask-guided cross-modal cross-attention mechanism, the network is encouraged to selectively attend to informative features during modality interaction, mitigating the risk of overfitting to the static distribution of the "Pseudo Ground Truth". Additionally, we propose a mask-guided and a task-coupled degradation-aware strategy to balance feature restoration and interaction. Extensive experiments on synthetic and real-world datasets (rain, haze, and snow) demonstrate that our method surpasses state-of-the-art approaches in visual quality, quantitative metrics, and downstream tasks.

## 1 INTRODUCTION

Multi-modality image fusion (MMIF) (Liu et al., 2024b; Zhang et al., 2021; Zhang & Demiris, 2023; Liu et al., 2024c; Jie et al., 2023) combines complementary information from multiple modalities to produce richer scene representations. For instance, in infrared and visible image fusion (IVIF), visible images capture texture details critical for semantic interpretation, while infrared images highlight salient targets via thermal radiation (Zhao et al., 2023a). By integrating these modalities, IVIF enhances downstream tasks like object detection (Wang et al., 2023), semantic segmentation (Liu et al., 2023), and depth estimation (Ranftl et al., 2020).

Existing studies (Li et al., 2023b; Zhao et al., 2024; Li et al., 2025b) have extensively investigated IVIF architectures, mainly focusing on ideal scenarios. Given a pair of registered infrared image ($IR$) and visible image ($VI$), the fusion task typically generates fused outputs by minimizing an L1-norm-based loss function. The optimization objective for conventional IVIF can be expressed as:

$$\theta^* = \arg\min_{\theta} \left( \lambda_1 \left\| f(VI, IR; \theta) - VI \right\|_1 + \lambda_2 \left\| f(VI, IR; \theta) - IR \right\|_1 \right) \tag{1}$$

Here, $f(\cdot)$ represents the fusion network, $\theta$ denotes the network parameters, and $\lambda_1$, $\lambda_2$ are weights balancing the contributions of both modalities. This objective enables the network to extract meaningful information from multiple modalities effectively. Some methods (Zhao et al., 2023a; Li et al., 2023b) augment this with additional loss functions to further optimize fusion performance.

However, real-world complexities like noise (Huang et al., 2024; Li et al., 2021), adverse weather (Li et al., 2024b), and motion blur (Chen et al., 2024) pose significant challenges to IVIF adaptability. Recent works (Li et al., 2024b; Zhang et al., 2024; Tang et al., 2024) addressing these conditions can be categorized into two types. The first follows a "restoration + fusion" paradigm, where

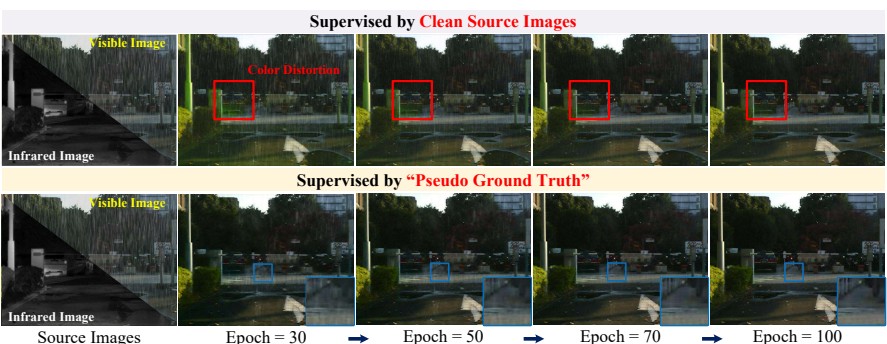

Figure 1: Visualization of the network (Li et al., 2023a) trained with different strategies.

each modality is first processed by a restoration network before being input into a fusion network. Nevertheless, this two-stage design introduces several issues: (1) Restoration focuses on intra-modal recovery, while fusion targets inter-modal complementarity, leading to unstable optimization. (2) Error accumulation: Artifacts from the restoration stage may propagate into the fusion process, amplifying degraded features and impairing reconstruction. To address these issues, some studies (Xu et al., 2024; Yi et al., 2024a; Jie et al., 2025; Sun et al., 2022; Rajasekaran et al., 2023) adopt a second strategy by using "Pseudo Ground Truth" as supervision. Specifically, the "Pseudo Ground Truth" refers to fusion results obtained by existing methods using clean $VI$ and $IR$ images as supervision targets. The optimization objective of this second strategy can be formulated as follows:

$$\theta^* = \arg\min_{\theta} \|f(VI, IR; \theta) - GT_{Pse}\|_1 \tag{2}$$

where $GT_{Pse}$ denotes the "Pseudo Ground Truth". As shown in Figure 1, we compare networks trained with source image supervision and "Pseudo Ground Truth" supervision. In the former, the network struggles to jointly perform multi-modal feature extraction and degradation removal, often leaving residual rain streaks and color distortions. In contrast, "Pseudo Ground Truth" supervision simplifies optimization, enabling the network to better capture global structures and fine details.

However, despite reducing the optimization complexity, this approach does not fully address fusion challenges in complex scenarios. Since "Pseudo Ground Truth" is derived from existing methods applied to clean images, it inherently carries information loss and modality bias. Moreover, networks trained with "Pseudo Ground Truth" often neglect complementary multi-modal features, limiting their generalization ability. For example, as shown in Figure 2, models trained with Pseudo Ground Truth supervision fail to effectively utilize thermal radiation information from infrared images in clean, degradation-free environments, leading to the loss of critical target regions. This highlights a key limitation: the strategy fits static distributions of pseudo targets rather than learning the dynamic mechanisms of IVIF. This raises a crucial question: *Can we harness the advantages of "Pseudo Ground Truth" while enabling the network to dynamically mine cross-modal information and remain degradation-aware?*

To address the aforementioned issues, we propose a mask-guided adverse weather image fusion framework (AMG-Fuse). Specifically, we first analyze the relationship between source images and fusion results, from which we derive a mask representation that decouples modality-specific components within the fused image. This mask enables effective separation of multi-modality features within the "Pseudo Ground Truth" and serves as an external prior to guide the network in explicitly modeling cross-modal interactions during training. This encourages the network to learn the dynamic fusion process. Furthermore, we design a Mask-Guided Feature Extraction Module (MFEM) that leverages the mask to restore degraded features while strengthening cross-modal feature interaction. During training, the proposed Mask-Guided Learning Strategy (MGLS) and Task-Coupled Degradation-Aware Learning Strategy (TDAS) collaboratively supervise the fusion network, facilitating the learning of clearer and more complementary feature representations. Our main contributions are as follows:

- We proposed a unified multi-modality image fusion method for complex scenarios, enabling simultaneous feature restoration and modality interaction within a unified network.

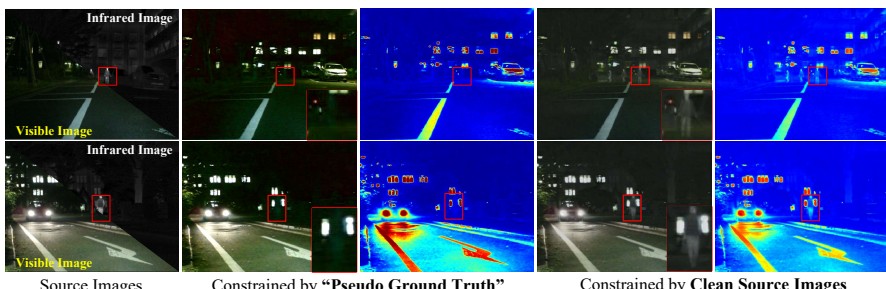

Figure 2: Visualization of the network (Li et al., 2023a) trained with different strategies.

- We proposed a **mask-guided feature extraction module** that effectively reconstructs the fusion features. Additionally, we designed a mask-guided network learning strategy to address the lack of dynamic cross-modal interaction modeling.
- We proposed a **task-coupled degradation-aware learning strategy** that leverages the restoration task as a meaningful supervisory signal to enhance degradation perception and guide the feature fusion process.
- We conducted extensive experiments under three adverse weather conditions to validate the effectiveness of the proposed method, and demonstrated its potential for real-world applications through downstream tasks.

## 2 RELATED WORKS

### 2.1 LEARNING BASED IVIF METHODS

Recent advancements in deep learning have significantly improved IVIF algorithms (Shi et al., 2025; Chen et al., 2025; Zhao et al., 2023b). Early works primarily focused on ideal conditions, emphasizing cross-modal information interaction (Shi et al., 2025; Zhao et al., 2023a; Li et al., 2023b; Liu et al., 2024a), unified fusion framework design(Xu et al., 2020; Cheng et al., 2025; Liang et al., 2022; Zhu et al., 2024), and joint optimization with downstream tasks (Tang et al., 2022a; Liu et al., 2022; Wu et al., 2025; Bai et al., 2024; Yang et al., 2025). Efficient cross-modal interaction relies on feature extractors (e.g., convolution or self-attention) to distinguish shared and modality-specific features, enabling unified perception and fusion. For example, (Zhao et al., 2023a) proposed a correlation-driven feature decomposition network using a dual-branch Transformer-CNN to separate global and local features, preserving salient details in fused outputs. With the increasing computational power and larger datasets, unified fusion frameworks have gained greater attention. (Cheng et al., 2025) leveraged digital photography tasks to guide the MMIF task and extended their proposed method to various fusion application scenarios. In addition, recent studies have increasingly focused on the application of fusion methods to downstream tasks. For example, (Wu et al., 2025) distilled the Segment Anything Model into the fusion network.

### 2.2 IVIF METHODS FOR COMPLEX SCENES

The complexity of real-world environments presents great challenges for IVIF algorithms in practical applications. Beyond fusion tasks in ideal conditions, researchers (Li et al., 2024a;b; Yi et al., 2024b; Tang et al., 2024; Li et al., 2025a) have shifted their focus to image fusion in complex conditions such as noise, overexposure, and adverse weather. The primary challenge in complex scenes is not only achieving cross-modal interaction, but also accurately distinguishing useful features from degradations and ensuring effective restoration. For example, (Yi et al., 2024b) proposed a degradation-aware interactive fusion algorithm guided by semantic text, using textual information as an external prior to help the network focus on existing degradations. (Tang et al., 2024) addressed fusion in complex scenarios by pretraining multiple degradation-robust conditional diffusion models for different modalities. (Li et al., 2024b) combined physical models and introduced an all-weather IVIF algorithm, incorporating various priors into the student model through distillation learning.

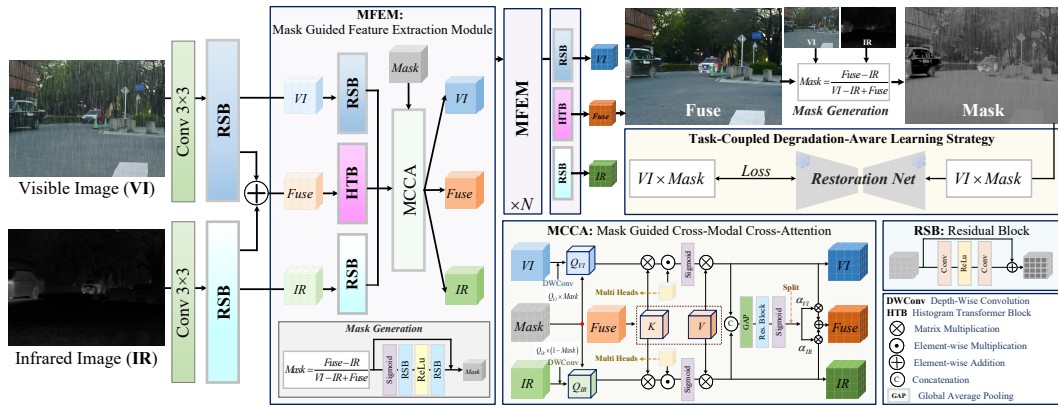

Figure 3: The flowchart of the proposed method.

However, although the above methods (Yi et al., 2024b; Tang et al., 2024; Li et al., 2024b) have advanced fusion research in complex scenarios, they tend to focus more on feature restoration and fail to thoroughly explore the paradigm differences between complex and ideal fusion.

## 3 PROPOSED METHOD

The workflow of the proposed algorithm is shown in Figure 3. Firstly, convolution and residual blocks are applied to enhance the feature dimensions of the input image. The extracted multi-modality features are then combined to produce the initial fused features. In the subsequent processing stage, residual blocks further refine the specific modal features, and the fused features are passed to the Histogram Transformer Block (HTB) (Sun et al., 2024) for prominent information extraction. The HTB segments spatial features based on pixel intensity and selectively processes features of different intensity ranges through a self-attention mechanism. This enables it to focus on similar degradation patterns over longer spatial distances (Sun et al., 2024). Finally, the extracted multi-modal and fused features are jointly refined using the Mask-Guided Cross-Modal Cross-Attention (MCCA) module, producing the fused image.

### 3.1 MASK CONSTRUCTION FROM FUSION FORMULATION

Although the introduction of "Pseudo Ground Truth" can ease the training process, it may also lead the network to directly replicate its features rather than learning the underlying modality allocation knowledge it conveys. To overcome this limitation, we decouple it by introducing the Mask. The core objective of the IVIF task is to suppress cross-modal redundancy and effectively extract complementary information. Disregarding feature enhancement and reduction, the fusion result ($Fuse$) can be expressed as:

$$Fuse = M \times VI + (1 - M) \times IR + \varepsilon \tag{3}$$

Here, $M$ denotes the mask for modality-specific weight allocation, and $\varepsilon$ represents the error and noise term. Given that $Fuse$, $VI$, and $IR$ are known, the expression for $M$ can be derived as:

$$M = \frac{Fuse - IR}{VI - IR} + \varepsilon \tag{4}$$

In real-world scenarios, directly subtracting infrared images from visible images can lead to misleading or unstable behaviour. Under haze or snow, visible images often exhibit excessive or locally overexposed brightness, whereas infrared images show reduced contrast and lower pixel values. After network optimisation and degradation removal, the fused output enhances the structural and target information from the infrared component, resulting in a more stable ($Fuse - IR$) distribution that better reflects the true scene brightness. However, because the brightness bias in the denominator of Equation 4 is driven by degenerative factor form visible image rather than semantic content, degraded information dominates the weighting, violating the decoupling principle of mask and preventing it

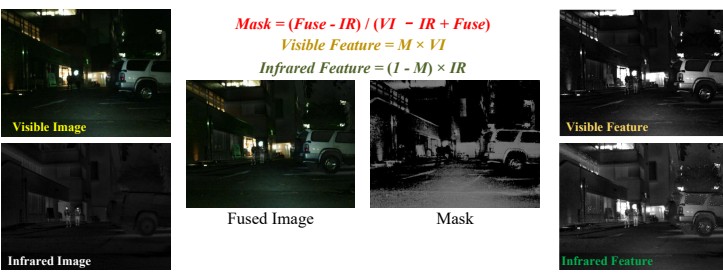

Figure 4: Visualization of the Mask and the Decoupled Multi-modality Features.

from capturing the actual modal distribution. Conversely, in night-time scenes, visible images contain minimal effective texture due to insufficient illumination, while infrared images remain relatively stable and may present stronger brightness cues. This causes $(VI - IR)$ to be negative or near zero over large regions, making Equation 4 numerically unstable when generating the mask. To address this issue, we rewrite the expression of $M$ as follows:

$$M = \frac{Fuse - IR}{VI - IR + Fuse} + \varepsilon \tag{5}$$

Incorporating the $Fuse$ information into the denominator effectively prevents the brightness bias of the visible image from erroneously amplifying the mask and avoids extreme numerical instability in the denominator. Finally, with $M$, we can then access the representation of different fusion algorithms for different modal information. As illustrated in Figure 4, we visualize the constructed mask and the decoupled multi-modal features. By guiding the network to learn the distribution pattern of multi-modal information in the "Pseudo Ground Truth", our method dynamically models the contribution of each modality, avoiding superficial fitting to the overall scene.

### 3.2 MASK GUIDED FEATURE EXTRACTION MODULE

Training directly on pseudo targets can lead the network to replicate their inherent biases, emphasizing superficial feature reproduction over effective cross-modal interactions. To address this, we propose the Mask-Guided Feature Extraction Module (MFEM).

The MFEM is designed based on Equation 3 and Equation 5. Firstly, multi-modality features are fed into the Residual Block (RSB), while the fused features are passed to the Histogram Transformer Block(HTB) for optimization. The resulting outputs are then used to compute the mask $M$. The MCCA module is proposed to enable the network to learn the multi-modal interaction patterns of the "Pseudo Ground Truth". The MCCA employs a cross-attention mechanism, where multi-modal features serve as queries guided by the mask, while the fused features act as keys and values.

To improve local information modeling, depthwise separable convolutions are introduced in both the query and key-value branches to expand the local receptive field. The Mask is then used to weight the queries $Q_{VI}$ and $Q_{IR}$, guiding the network to focus on the significant feature in each modality. This enables decoupling and recombination of multi-modal features within the fusion space.

### 3.3 MASK GUIDED LEARNING STRATEGY

To better leverage "Pseudo Ground Truth" for learning multi-modal interaction features, we propose a Mask Guided Learning Strategy (MGLS). Specifically, the mask $M_{Pse}$ is computed based on the "Pseudo Ground Truth" ($GT_{Pse}$) and the clean multi-modality source images $VI_C$ and $IR_C$, as shown in Equation 5. This mask is then used to decouple the multi-modality features $F_{VI}$ and $F_{IR}$ in the "Pseudo Ground Truth".

$$F_{VI} = M_{Pse} \times VI_C, \quad F_{IR} = (1 - M_{Pse}) \times IR_C \tag{6}$$

Here, $F_{VI}$ and $F_{IR}$ are the modality allocation maps in $GT_{Pse}$. Then, we compute the L1 loss for the corresponding multi-modal features to constrain the network in learning the multi-modal information distribution pattern in the "Pseudo Ground Truth". Let the network output the multi-modal features

| Infrared Images | Visible Images | Fused Image (**In Clean**) | Masks | **Visible Image × Mask** |

Figure 5: Visual analysis of the relevant components in Equation 8.

as $\hat{F}_{VI}$ and $\hat{F}_{IR}$, the proposed MGLS loss ($\mathcal{L}_{MGLS}$) is computed as follows:

$$\mathcal{L}_{MGLS} = \frac{1}{HW} \left( \|\hat{F}_{VI} - F_{VI}\|_1 + \|\hat{F}_{IR} - F_{IR}\|_1 \right) \tag{7}$$

where $H$ and $W$ represent the length and width of the image respectively.

## 3.4 TASK-COUPLED DEGRADATION-AWARE LEARNING STRATEGY

The Equation 3 defines the synthesis paradigm of conventional image fusion. Therefore, after determining the mask, the allocation strategies of the existing methods for different modal information can be indirectly obtained. In complex scenes, the two input images ($VI_{Deg}$ and $IR_{Deg}$) are often degraded. At this point, the goal of the fusion network is to output a clear fusion result that contains complementary multi-modal features. This process can be modeled analogously using Equation 3.

$$Fuse = M_{Deg} \times VI_{Deg} + (1 - M_{Deg}) \times IR_{Deg} + \epsilon \tag{8}$$

Since $Fuse$ is a non-degraded image, the visible modal information obtained by $M_{Deg} \times VI_{Deg}$ in Equation 8 should be degradation-free. Thus, the mask can effectively capture the distribution of the degraded area. This is demonstrated in figure 5, which shows that most of the rain streaks have been successfully suppressed or removed. Based on the above observation results, we propose the Task-Coupled Degradation Perceptual Learning Strategy (TDAS), aiming to guide the fusion network to prioritize processing clearer and more prominent regions. For the restoration task, when the input image is degradation-free, the restoration network tends to generate an output that closely resembles the original image. Therefore, we define the TDAS loss as follows:

$$\mathcal{L}_{\text{TDAS}} = \frac{1}{H \times W} \|VI_F - \mathcal{R}(VI_F)\|_1 \tag{9}$$

where $\mathcal{R}(\cdot)$ represents the restoration model (Zamir et al., 2022), and $VI_F = M_{Deg} \times VI_{Deg}$. When $VI_F$ is similar enough to the output $\mathcal{R}(VI_F)$ of the restoration network, it means that the fusion network has effectively restored the clear features.

## 3.5 LOSS FUNCTIONS

During training, in addition to MGLS and TDAS, source image supervision is further introduced to enhance the ability of the model to capture multi-modal distributions by imposing constraints on the fusion output. Specifically, we adopt a gradient loss and a color consistency loss to preserve structural details and color distribution consistency, respectively. The gradient loss is defined as follows:

$$\mathcal{L}_{\text{grad}} = \frac{1}{H \times W} \|\nabla Fuse - \max(\nabla VI_C, \nabla IR_C)\|_1 \tag{10}$$

The color consistency loss (Yi et al., 2024b) is expressed as follows,

$$\mathcal{L}_{\text{color}} = \frac{1}{H \times W} \|F_{\text{CbCr}}(Fuse) - F_{\text{CbCr}}(VI_C)\|_1 \tag{11}$$

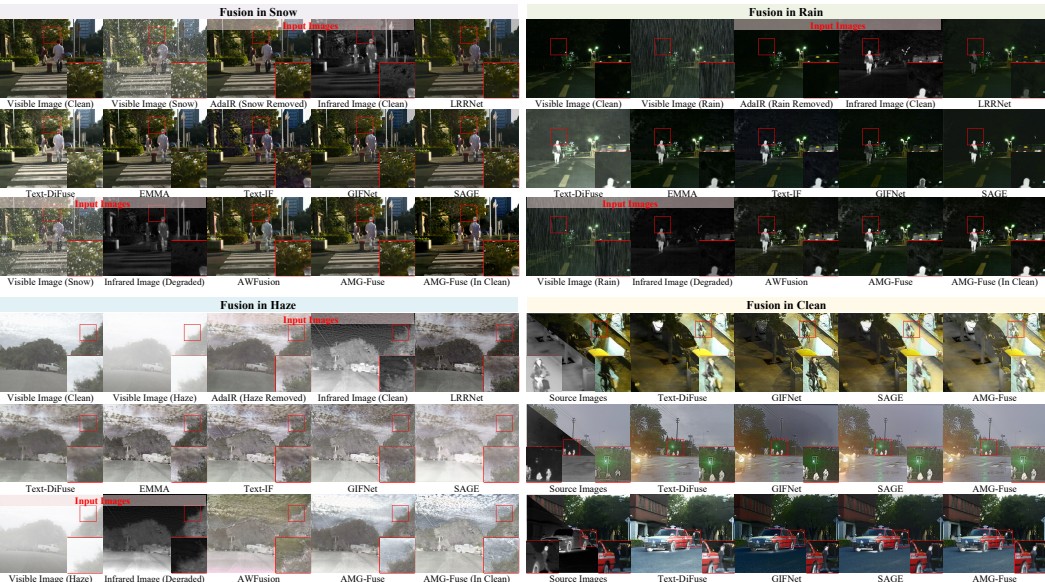

Figure 6: Qualitative comparison results of all methods in the four scenarios.

where $F_{\text{CbCr}}$ represents the function of transferring the image space to the CbCr color space. The total loss function is expressed as follows,

$$\mathcal{L}_{\text{total}} = \lambda \times \mathcal{L}_{\text{MGLS}} + \mathcal{L}_{\text{TDAS}} + \mathcal{L}_{\text{color}} + \mathcal{L}_{\text{grad}} \tag{12}$$

where $\lambda$ is the decay coefficient, which gradually decreases as the number of training epochs increases. All loss terms except $\mathcal{L}_{\text{MGLS}}$ are assigned a weight of 1, ensuring balanced contributions from gradient and color consistency constraints without requiring careful hyperparameter tuning.

## 4 EXPERIMENTS

### 4.1 EXPERIMENTAL SETUPS

**Datasets:** Our experiments are conducted in two scenarios: adverse weather and ideal environments. For the adverse weather experiments, we selected 1,000 images from each of the three distinct weather conditions (Snow, Rain, Haze) in the AWMM-100k dataset (Li et al., 2024b) for training, and 150 images for testing. In addition, we evaluated our method on 23 real degraded images from AWMM-100k to further validate its performance under authentic degradation. For the ideal environment experiments, we utilized three public datasets: M3FD (Liu et al., 2022), MSRS (Tang et al., 2022b), and LLVIP (Jia et al., 2021).

Notably, rain and snow degraded images in AWMM-100k are generated from masks extracted from real video sequences and rendered to simulate realistic degradation. Training on these images equips the model with the ability to robustly process real-world degraded scenes.

**Comparative Methods:** We adopted the "restoration+ fusion" scheme for performance comparison. The adaptive image restoration network (AdaIR) (Cui et al., 2025) was employed as the restoration module and retrained on the AWMM-100k dataset. The fusion methods include LRRNet (Li et al., 2023a), Text-DiFuse (Zhang et al., 2024), EMMA (Zhao et al., 2024), Text-IF (Yi et al., 2024b), GIFNet (Cheng et al., 2025), SAGE (Wu et al., 2025), and AWFusion (Li et al., 2024b). Among these, only Text-DiFuse, Text-IF and AWFusion are specifically designed for complex scenes. However, as Text-DiFuse and Text-IF lacks the capability to handle adverse weather conditions, it is categorized as a standard fusion method. All methods were trained on the same datasets used in our experiments for fair evaluation. A detailed discussion of this comparison strategy is provided in A.2.

**Metrics:** We evaluated the performance of different image fusion methods from multiple perspectives using five types of metrics: (1) Information Theory-Based: Normalized Mutual Information ($Q_{MI}$);

Table 1: Quantitative comparison of all methods in three adverse weather scenarios. The marked red indicates the best score and the marked green indicates the second score.

| Source | Methods | Pub. | Restoration | $Q_{MI}$ | $Q_G$ | $Q_M$ | $VIF$ | $Q_{CB}$ | $SSIM$ | $Q^{AB/F}$ |
|---|---|---|---|---|---|---|---|---|---|---|
| Snow | LRRNet | PAMI 23 | AdaIR | 0.4403 | 0.2849 | 0.3779 | 0.2588 | 0.3661 | 0.1566 | 0.3970 |
| | Text-DiFuse | NIPS 24 | | 0.4087 | 0.3047 | 0.3633 | 0.2433 | 0.4264 | 0.2821 | 0.4130 |
| | EMMA | CVPR 24 | | 0.5335 | 0.4235 | 0.5353 | 0.3252 | 0.5011 | 0.4096 | 0.5470 |
| | Text-IF | CVPR 24 | | 0.4399 | 0.4199 | 0.5942 | 0.3300 | 0.4929 | 0.4298 | 0.5419 |
| | GIFNet | CVPR 25 | | 0.3388 | 0.3282 | 0.3305 | 0.2732 | 0.4854 | 0.3722 | 0.3882 |
| | SAGE | CVPR 25 | | 0.4693 | 0.3149 | 0.4752 | 0.3819 | 0.4421 | 0.3474 | 0.4585 |
| | AWFusion | Arxiv 24 | x | 0.4668 | 0.3552 | 0.5535 | 0.3024 | 0.4798 | 0.3734 | 0.4937 |
| | AMG-Fuse | – | | 0.4960 | 0.4240 | 0.6408 | 0.3463 | 0.5062 | 0.4302 | 0.5519 |
| Rain | LRRNet | PAMI 23 | AdaIR | 0.4002 | 0.2306 | 0.4119 | 0.2287 | 0.3335 | 0.0453 | 0.3334 |
| | Text-DiFuse | NIPS 24 | | 0.3546 | 0.2442 | 0.3645 | 0.2320 | 0.3909 | 0.1962 | 0.3329 |
| | EMMA | CVPR 24 | | 0.4536 | 0.3745 | 0.5871 | 0.3384 | 0.4785 | 0.3700 | 0.4866 |
| | Text-IF | CVPR 24 | | 0.3717 | 0.4085 | 0.6375 | 0.3489 | 0.4824 | 0.4046 | 0.5149 |
| | GIFNet | CVPR 25 | | 0.2886 | 0.2975 | 0.3686 | 0.3091 | 0.4834 | 0.3474 | 0.3544 |
| | SAGE | CVPR 25 | | 0.3994 | 0.2286 | 0.5086 | 0.3478 | 0.3964 | 0.2734 | 0.3670 |
| | AWFusion | Arxiv 24 | x | 0.3509 | 0.3334 | 0.6428 | 0.2997 | 0.4691 | 0.3528 | 0.4618 |
| | AMG-Fuse | – | | 0.4084 | 0.4163 | 0.6940 | 0.3582 | 0.4879 | 0.4080 | 0.5184 |
| Haze | LRRNet | PAMI 23 | AdaIR | 0.2411 | 0.3044 | 0.3531 | 0.2496 | 0.4306 | 0.2031 | 0.3701 |
| | Text-DiFuse | NIPS 24 | | 0.2304 | 0.3137 | 0.3463 | 0.2234 | 0.4391 | 0.3035 | 0.4021 |
| | EMMA | CVPR 24 | | 0.2345 | 0.3553 | 0.4149 | 0.2559 | 0.4567 | 0.3296 | 0.4617 |
| | Text-IF | CVPR 24 | | 0.2312 | 0.3999 | 0.4418 | 0.2779 | 0.4646 | 0.3443 | 0.4883 |
| | GIFNet | CVPR 25 | | 0.2451 | 0.3124 | 0.3181 | 0.2401 | 0.4784 | 0.3463 | 0.3478 |
| | SAGE | CVPR 25 | | 0.2312 | 0.3416 | 0.3948 | 0.2850 | 0.4553 | 0.3316 | 0.4336 |
| | AWFusion | Arxiv 24 | x | 0.2639 | 0.4283 | 0.5471 | 0.3433 | 0.4610 | 0.3901 | 0.5322 |
| | AMG-Fuse | – | | 0.3096 | 0.4376 | 0.5438 | 0.3445 | 0.4864 | 0.4083 | 0.5414 |

Table 2: Quantitative comparison of all methods in real-world scenarios. The marked red indicates the best score and the marked green indicates the second score.

| Methods | Pub. | $Q_{MI}$ | $Q_G$ | $Q_M$ | $VIF$ | $Q_{CB}$ | $SSIM$ | $Q^{AB/F}$ |
|---|---|---|---|---|---|---|---|---|
| LRRNet | PAMI 23 | 0.5429 | 0.3944 | 0.9170 | 0.4079 | 0.4694 | 0.4174 | 0.4773 |
| Text-DiFuse | NIPS 24 | 0.5174 | 0.3325 | 0.9384 | 0.3850 | 0.4108 | 0.3848 | 0.4657 |
| EMMA | CVPR 24 | 0.5497 | 0.3884 | 0.8056 | 0.3264 | 0.4470 | 0.4342 | 0.5301 |
| Text-IF | CVPR 24 | 0.5258 | 0.5748 | 1.0852 | 0.3566 | 0.4936 | 0.4972 | 0.6800 |
| GIFNet | CVPR 25 | 0.5085 | 0.3673 | 0.8833 | 0.3021 | 0.4250 | 0.4489 | 0.4605 |
| SAGE | CVPR 25 | 0.5200 | 0.4786 | 1.0538 | 0.3647 | 0.4298 | 0.4816 | 0.5314 |
| AWFusion | Arxiv 24 | 0.5032 | 0.4547 | 1.0028 | 0.3523 | 0.4649 | 0.5036 | 0.5860 |
| AMG-Fuse | – | 0.7234 | 0.5114 | 1.1022 | 0.4373 | 0.4786 | 0.5202 | 0.5957 |

(2) Feature-Based: Gradient-Based Fusion Performance ($Q_G$) and Multiscale Image Fusion Metric ($Q_M$); (3) Structural Similarity-Based: Structural Similarity Index Measure ($SSIM$); (4) Human Perception Inspired: Chen-Blum Metric ($Q_{CB}$) and Visual Information Fidelity ($VIF$); and (5) Other Comprehensive Metrics: Normalized Weighted Performance Metric ($Q^{AB/F}$) (Liu et al., 2011). For all metrics, a higher value indicates better image quality.

**Training details:** During the training process, we randomly cropped images from the AWMM-100k training set into $168 \times 168$ patches, the training runs for 200 epochs. The Adam optimizer was used with an initial learning rate of $1 \times 10^{-3}$ and a batch size of 2. All the experiments were conducted in the PyTorch 2.1.1 environment, and the server was equipped with four GeForce RTX 3090 GPUs. The generation of "Pseudo Ground Truth" can be found in the A.3.

## 4.2 QUALITATIVE COMPARISON

Figure 6 presents fusion results under three types of adverse weather conditions. For most fusion methods, the visible input is first restored by AdaIR before fusion. AMG-Fuse (Clean) represents the fusion result obtained using clean source images as input, serving as an upper-bound reference.

**Snow Weather:** While AdaIR effectively removes snow interference, it causes significant detail loss, resulting in blurred output for LRRNet, Text-IF, and SAGE. In contrast, our method (AMG-Fuse) ef-

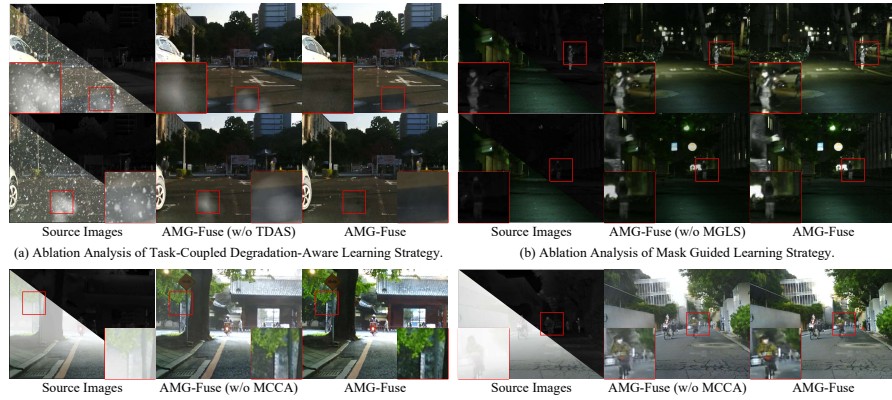

Source Images    AMG-Fuse (w/o TDAS)    AMG-Fuse      Source Images    AMG-Fuse (w/o MGLS)    AMG-Fuse

(a) Ablation Analysis of Task-Coupled Degradation-Aware Learning Strategy.    (b) Ablation Analysis of Mask Guided Learning Strategy.

Source Images    AMG-Fuse (w/o MCCA)    AMG-Fuse      Source Images    AMG-Fuse (w/o MCCA)    AMG-Fuse

(c) Ablation Analysis of Mask Guided Cross-Modal Cross-Attention.

Figure 7: Visual analysis of ablation studies: (a) TDAS component ablation; (b) MGLS component ablation; (c) MCCA component ablation.

fectively eliminates degradation artifacts, preserves key details, and fully exploits the complementary information between modalities.

**Rain Weather:** AdaIR removes rain streaks but oversmooths textures, leading to diminished scene details. LRRNet and GIFNet fail to sufficiently highlight infrared cues, while EMMA, Text-IF, and SAGE enhance occluded visible targets but fail to capture fine infrared textures. AMG-Fuse, however, maintains a strong balance between multi-modal interaction and subtle visible details.

**Haze Weather:** Haze introduces depth-dependent degradation, complicating feature extraction. LRRNet, Text-DiFuse, and SAGE exhibit significant contrast loss, while EMMA and Text-IF produce outputs that diverge from perceptual expectations. AWFusion introduces color distortions, further degrading visual fidelity. In contrast, AMG-Fuse effectively suppresses haze artifacts, generating visually natural and structurally rich fused images.

**Fusion in Clean Scene** As shown in Figure 6, we compare AMG-Fuse with three representative fusion methods in clean environments. Despite being designed for degraded scenarios, AMG-Fuse demonstrates robust multi-modality feature extraction and produces competitive results in ideal conditions, showcasing its adaptability and generalization capability.

### 4.3 QUANTITATIVE COMPARISON

Table 1 presents the quantitative results of AMG-Fuse and competing methods under the three adverse weather conditions. AMG-Fuse consistently ranks in the top two across all metrics and scenarios, outperforming other methods in several key metrics. Specifically, AMG-Fuse improves 3.67%, 3.86% and 3.56% over the sub-optimal method in snow, rain, and haze scenes, respectively. The quantitative comparison in the standard dataset can be found in the A.5.

To further validate the applicability of the proposed algorithm on real-world data, we conducted experiments on the AWMM-100k real haze dataset, with quantitative results shown in Table 2. The proposed method achieves top-two scores across all metrics, demonstrating the effectiveness of the mask-guided learning strategy. This strategy prevents the model from merely learning static feature recovery, enabling it to restore fine details while facilitating multi-modality information interaction.

### 4.4 ABLATION STUDIES

The following presents a qualitative analysis of our ablation study, while the corresponding quantitative results are provided in the A.6.

**Analysis of Task-Coupled Degradation-Aware Learning Strategy:** As shown in Figure 7(a), removing TDAS (AMG-Fuse w/o TDAS) significantly reduces fusion performance, particularly under severe degradation, where structural details are harder to recover. TDAS is crucial for improving

Table 3: Detection accuracy comparison on the M3FD dataset.

| Methods | Pub. | People | Car | Bus | Lamp | Motorcycle | Truck | mAP@0.5 | mAP@[0.5:0.95] |
|---|---|---|---|---|---|---|---|---|---|
| LRRNet | *PAMI 23* | 0.785 | 0.909 | 0.912 | 0.764 | 0.721 | 0.808 | 0.817 | 0.484 |
| Text-DiFuse | *NIPS 24* | 0.79 | 0.91 | 0.927 | 0.793 | 0.69 | 0.795 | 0.817 | 0.52 |
| EMMA | *CVPR 24* | 0.784 | 0.904 | 0.89 | 0.722 | 0.69 | 0.78 | 0.795 | 0.501 |
| Text-IF | *CVPR 24* | 0.812 | 0.92 | 0.923 | 0.806 | 0.744 | 0.811 | 0.836 | 0.535 |
| GIFNet | *CVPR 25* | 0.806 | 0.906 | 0.919 | 0.777 | 0.671 | 0.779 | 0.81 | 0.506 |
| SAGE | *CVPR 25* | 0.814 | 0.919 | 0.921 | 0.815 | 0.698 | 0.83 | 0.833 | 0.534 |
| AWFusion | *Arxiv 24* | 0.791 | 0.922 | 0.931 | 0.804 | 0.726 | 0.821 | 0.832 | 0.534 |
| AMG-Fuse | – | 0.837 | 0.931 | 0.934 | 0.816 | 0.728 | 0.813 | 0.843 | 0.541 |

fusion quality by constraining the training process using the restoration model's ability to preserve clean image features.

**Analysis of Mask Guided Learning Strategy:** As depicted in Figure 7(b), removing MGLS (AMG-Fuse w/o MGLS) causes the fusion network to rely excessively on the static pixel distribution of the "Pseudo Ground Truth." Without MGLS, the network lacks sufficient guidance for dynamic modal feature assignment, resulting in suboptimal performance and reduced adaptability to complex scenes.

**Analysis of Mask Guided Cross-Modal Cross-Attention:** As shown in Figure 7(c), without MCCA (AMG-Fuse w/o MCCA), the network becomes a unidirectional recovery structure, limiting multi-modal integration and leading to inferior fusion results.

### 4.5 DOWNSTREAM TASK EXPERIMENT

We evaluated the practicality of AMG-Fuse in downstream tasks, we conducted object detection experiments on the M3FD dataset using YOLOv7 (Wang et al., 2023). Table 3 shows that our proposed method ranks top two across all target categories and achieves the highest scores for mAP@0.5 and mAP@[0.5:0.95]. Moreover, Figure 8 provides a qualitative comparison of different algorithms on the M3FD dataset. The proposed method accurately detects all object information in the scene and achieves high detection reliability. In contrast, Text-IF exhibits detection errors due to the introduction of incorrect semantic cues, while the low-contrast outputs of GIFNet further reduce its detection precision. Overall, the proposed algorithm improves the utility of fused images for downstream tasks.

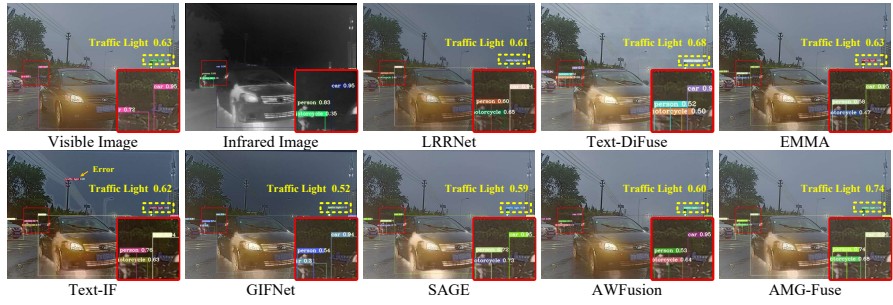

Figure 8: Qualitative comparison of all methods on object detection tasks.

### 5 CONCLUSION

In this paper, we propose a mask-guided multi-modality image fusion method capable of addressing the fusion problem in both severe weather and ideal scenarios. We introduced "Pseudo Ground Truth" to simplify training and derived a mask representation to guide the network in learning dynamic cross-modal interactions. To further enhance feature representation, we designed the Mask-Guided Learning Strategy and Task-Coupled Degradation-Aware Learning Strategy, which improve both feature restoration and interaction. Extensive experiments demonstrate that AMG-Fuse consistently outperforms state-of-the-art methods across multiple scenarios.

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

# A  APPENDIX

## A.1  USE OF LLMS

Large Language Models (LLMs) were used to aid in the writing and polishing of the manuscript. Specifically, we used an LLM to assist in refining the language, improving readability, and ensuring clarity in various sections of the paper. The model helped with tasks such as sentence rephrasing. grammar checking, and enhancing the overall flow of the text.

It is important to note that the LLM was not involved in the ideation, research methodology, or experimental design. All research concepts, ideas, and analyses were developed and conducted by the authors. The contributions of the LLM were solely focused on improving the linguistic quality of the paper, with no involvement in the scientific content or data analysis.

The authors take full responsibility for the content of the manuscript, including any text generated or polished by the LLM. We have ensured that the LLM-generated text adheres to ethical guidelines and does not contribute to plagiarism or scientific misconduct.

## A.2  DISCUSSION OF COMPARISON STRATEGIES

In the task of multi-modal image fusion under adverse weather conditions, a reasonable comparison strategy is crucial for fairly and objectively evaluating the performance of the proposed algorithm. We mainly consider the following four strategies:

1. "Image Restoration + Image Fusion" vs "Image Restoration + Proposed Method";
2. "Image Fusion (with clean images)" vs "Proposed Method (with degraded images)";
3. "Image Restoration + Image Fusion" vs "Proposed Method (with degraded images)";
4. "Image Fusion (with degraded images)" vs "Proposed Method (with degraded images)".

For strategy (1), one of the key objectives of our unified framework is to address the problem of error accumulation inherent in the traditional two-stage pipeline (first restoration and then fusion). If we adopt the form of "Image Restoration + Proposed Method," the artifacts and errors introduced during the restoration stage would be propagated into our model. This not only contradicts the motivation of our framework but also results in an unfair comparison.

For strategy (2), most existing fusion algorithms are incapable of handling degraded images under adverse weather, and their best-case input is a clean source image. However, comparing them directly with our method, which processes degraded inputs, creates a fundamentally unequal starting condition. Hence, this strategy lacks fairness.

For strategy (3), this is currently the most reasonable and fair comparison. On the one hand, it aligns with the motivation of our unified model to avoid the drawbacks of two-stage processing. On the other hand, both the baselines and the proposed method use degraded images as inputs, ensuring consistency in experimental conditions and thereby providing more convincing results.

For strategy (4), this is the ideal way to compare our approach with existing fusion methods that are explicitly designed for adverse weather conditions. Since the amount of input information is identical and the compared methods are also trained on adverse-weather datasets, this strategy allows for a fair and precise evaluation of our method's advantages over current degradation-aware fusion approaches.

It should be noted that the performance of existing comparison methods is essentially limited by the effectiveness of their preceding restoration modules (e.g., AdaIR). Therefore, comparisons conducted only on adverse-weather datasets may not fully reflect the potential of our method. To address this, we additionally performed experiments on standard multi-modal fusion datasets such as MSRS, M3FD, and LLVIP, where the source images are free of degradation. These experiments further validate the effectiveness and generalization capability of our method in ideal scenarios.

## A.3  THE DISCUSSION AND GENERATION OF "PSEUDO GROUND TRUTH"

"Pseudo Ground Truth" serves as a guidance signal that helps the model learn modality allocation behavior. Two natural concerns arise: (1) whether introducing "Pseudo Ground Truth" might cap

the performance upper bound of the proposed method, and (2) whether it may propagate inherent modality bias into the fusion process. To address these issues, we provide a detailed conceptual analysis followed by empirical validation. The analysis is summarized as follows:

- The proposed algorithm does not rely entirely on the knowledge provided by "Pseudo Ground Truth" in the loss function. "Pseudo Ground Truth" serve only as a soft supervisory signal rather than the sole supervision. Clean source images are also employed as standard supervision to encourage the network to learn the most effective modality allocation strategy.

- "Pseudo Ground Truth" are generated from clean source images and thus contain relatively high signal-to-noise structural and textural priors. Even if the modality allocation in "Pseudo Ground Truth" is biased, the effective information density at the pixel level is typically higher than that of inputs degraded by adverse weather. Therefore, the role of "Pseudo Ground Truth" is closer to a learnable beneficial prior, which helps narrow the search space and stabilize early optimization, rather than imposing a strict limit on the ultimate performance of model.

- The loss weight of the Modality-Guided Learning Strategy (MGLS) gradually decays during training: it is relatively high in the early stage to stabilize optimization and accelerate convergence, and gradually diminishes later so that the model mainly relies on the supervision from clean source images to learn the optimal fusion strategy. This mechanism ensures that "Pseudo Ground Truth" function as an early-stage "teacher guide" without constraining the model from surpassing their performance in later learning.

While the above analysis suggests that "Pseudo Ground Truth" do not restrict the model's capacity, we further validate this claim empirically. Specifically, we first used EMMA to generate fused results on clean multi-modal images and treated them as "Pseudo Ground Truth". Then, we fed the same clean multi-modal inputs into our proposed model and directly compared the fusion outputs of both methods. According to the hypothesis, if our model were strictly limited by the modality distribution and interaction patterns provided by EMMA during training, then the performance of EMMA should represent the upper bound of our model. However, as shown in the Table 4, our method significantly surpasses EMMA across all three standard datasets. This evidence directly refutes the concern that "Pseudo Ground Truth" could restrict the upper-bound performance of model, demonstrating that our model is not constrained by the representational capacity of the "Pseudo Ground Truth". Instead, it can further refine feature interactions and fusion strategies beyond the priors encoded in EMMA.

In summary, the goal of using "Pseudo Ground Truth" is to provide the network with a more reliable optimization trajectory during the early training stage. We adopt EMMA as the generator, not to seek the optimal "Pseudo Ground Truth", but to supply a reliable prior for facilitating convergence. Since EMMA has already been trained on existing multi-modal datasets, we fine-tune it on the clean subset of AWMM-100k comprising 1,000 images, while preserving its original weights. All training configurations strictly follow those reported in the original paper.

Table 4: Quantitative comparison between the proposed method and the "Pseudo Ground Truth" generator EMMA on three datasets. The marked red indicates the best score.

| Source | Methods | Pub. | $Q_{MI}$ | $Q_G$ | $Q_M$ | $VIF$ | $Q_{CB}$ | $SSIM$ | $Q^{AB/F}$ |
|--------|---------|------|----------|-------|-------|-------|----------|--------|------------|
| MSRS | EMMA | *CVPR 24* | 0.6697 | 0.5624 | 0.7291 | 0.4165 | 0.5394 | 0.4746 | 0.6500 |
| | AMG-Fuse | _ | 0.7594 | 0.5952 | 1.5051 | 0.4706 | 0.5809 | 0.5044 | 0.6749 |
| M3FD | EMMA | *CVPR 24* | 0.5646 | 0.4827 | 0.5595 | 0.3829 | 0.4880 | 0.4787 | 0.6032 |
| | AMG-Fuse | _ | 0.7010 | 0.5376 | 1.2994 | 0.4713 | 0.4979 | 0.5069 | 0.6384 |
| LLVIP | EMMA | *CVPR 24* | 0.4437 | 0.5005 | 0.2314 | 0.3301 | 0.4559 | 0.4339 | 0.5969 |
| | AMG-Fuse | _ | 0.4684 | 0.6228 | 0.7938 | 0.3952 | 0.4479 | 0.4574 | 0.6907 |

## A.4   MORE DETAILS ABOUT EXPERIMENTAL SETUP

The proposed algorithm uses an autoencoder structure, based on MFEM as the feature extraction module, and constructs a UNet architecture consisting of three encoding and three decoding layers.

At each stage, the number of HTB and RSB is set to $\{4, 4, 6, 8\}$, while the number of self-attention heads in HTB and MCCA is set to $\{2, 4, 6, 8\}$, respectively.

## A.5 QUANTITATIVE COMPARISON RESULTS IN STANDARD DATASETS

To validate the multi-modality feature extraction and fusion capability of the proposed algorithm in an ideal environment, we conducted experiments on three standard datasets, MSRS, M3FD, and LLVIP. Table 5 reports the quantitative evaluation results of each comparison method on the above datasets. It can be observed that the proposed method achieves the best results in all the evaluation metrics on the MSRS dataset; on the M3FD and LLVIP datasets, it ranks first in 5 and 6 metrics, respectively. These results further validate the superior performance and strong generalization ability of the proposed algorithm in the multi-modality image fusion task.

Table 5: Quantitative comparison of all methods in three standard datasets (MSRS, M3FD, LLVIP). The marked red indicates the best score ,and the marked green indicates the second score.

| Datasets | Methods | Pub. | $Q_{MI}$ | $Q_G$ | $Q_M$ | $VIF$ | $Q_{CB}$ | $SSIM$ | $Q^{AB/F}$ |
|---|---|---|---|---|---|---|---|---|---|
| MSRS | LRRNet | PAMI 23 | 0.4825 | 0.3714 | 0.4573 | 0.4152 | 0.3418 | 0.2546 | 0.4797 |
| | Text-DiFuse | NIPS 24 | 0.4357 | 0.3732 | 0.4145 | 0.4690 | 0.2864 | 0.3448 | 0.4753 |
| | EMMA | CVPR 24 | 0.6697 | 0.5624 | 0.7291 | 0.4165 | 0.5394 | 0.4746 | 0.6500 |
| | Text-IF | CVPR 24 | 0.4825 | 0.5698 | 0.8709 | 0.3976 | 0.5053 | 0.4749 | 0.6666 |
| | GIFNet | CVPR 25 | 0.3190 | 0.3816 | 0.3834 | 0.2858 | 0.5085 | 0.4239 | 0.4334 |
| | SAGE | CVPR 25 | 0.5363 | 0.4842 | 0.7171 | 0.4475 | 0.5013 | 0.4582 | 0.5824 |
| | AWFusion | Arxiv 24 | 0.4587 | 0.4880 | 0.6728 | 0.3719 | 0.5082 | 0.4601 | 0.5994 |
| | AMG-Fuse | – | 0.7594 | 0.5952 | 1.5051 | 0.4706 | 0.5809 | 0.5044 | 0.6749 |
| M3FD | LRRNet | PAMI 23 | 0.3871 | 0.3844 | 0.4487 | 0.3900 | 0.4545 | 0.3876 | 0.5102 |
| | Text-DiFuse | NIPS 24 | 0.3335 | 0.1631 | 0.2453 | 0.1026 | 0.3786 | 0.0835 | 0.1902 |
| | EMMA | CVPR 24 | 0.5646 | 0.4827 | 0.5595 | 0.3829 | 0.4880 | 0.4787 | 0.6032 |
| | Text-IF | CVPR 24 | 0.3875 | 0.5013 | 0.7665 | 0.3432 | 0.5043 | 0.4496 | 0.6544 |
| | GIFNet | CVPR 25 | 0.3804 | 0.4161 | 0.4041 | 0.2918 | 0.4795 | 0.4710 | 0.5243 |
| | SAGE | CVPR 25 | 0.4241 | 0.4765 | 0.7106 | 0.4110 | 0.4770 | 0.4909 | 0.5974 |
| | AWFusion | Arxiv 24 | 0.4179 | 0.4460 | 0.6359 | 0.3609 | 0.4769 | 0.4654 | 0.5940 |
| | AMG-Fuse | – | 0.7010 | 0.5376 | 1.2994 | 0.4713 | 0.4979 | 0.5069 | 0.6384 |
| LLVIP | LRRNet | PAMI 23 | 0.4048 | 0.3327 | 0.1629 | 0.3807 | 0.4237 | 0.3523 | 0.4232 |
| | Text-DiFuse | NIPS 24 | 0.4041 | 0.3159 | 0.1723 | 0.2691 | 0.4764 | 0.3411 | 0.4328 |
| | EMMA | CVPR 24 | 0.4437 | 0.5005 | 0.2314 | 0.3301 | 0.4559 | 0.4339 | 0.5969 |
| | Text-IF | CVPR 24 | 0.3931 | 0.5498 | 0.3331 | 0.3227 | 0.5142 | 0.4463 | 0.6624 |
| | GIFNet | CVPR 25 | 0.3660 | 0.3813 | 0.1533 | 0.2126 | 0.5129 | 0.3855 | 0.4515 |
| | SAGE | CVPR 25 | 0.4067 | 0.4786 | 0.2748 | 0.3590 | 0.4573 | 0.4223 | 0.5631 |
| | AWFusion | Arxiv 24 | 0.4646 | 0.5684 | 0.4463 | 0.3947 | 0.5315 | 0.4560 | 0.6612 |
| | AMG-Fuse | – | 0.4684 | 0.6228 | 0.7938 | 0.3952 | 0.4479 | 0.4574 | 0.6907 |

## A.6 QUANTITATIVE COMPARISON RESULTS IN ABLATION STUDIES

**Analysis of Task-Coupled Degradation-Aware Learning Strategy:** When this strategy (i.e., AMG-Fuse (w/o TDAS)) is removed, faced with the presence of significant degradation in the visible image, it is no longer difficult to reconstruct a clear structure relying only on multimodal feature extraction, resulting in a significant degradation in the perceptual quality of the fused image. This phenomenon is further verified by the quantitative results in Table 6, where all the evaluation metrics show different degrees of degradation, indicating that TDAS is of great value in enhancing model robustness.

**Analysis of Mask Guided Learning Strategy:** As shown in Table 6, under severe weather conditions, all metrics decreased after the removal of MGLS. While the degradation is limited in the human vision-based metrics (e.g., $VIF$ and $Q_{CB}$) due to the presence of "Pseudo Ground Truth", the degradation is significant in the more feature structure-dependent metrics (e.g., $Q_G$ and $Q_M$), with the mean value of all metrics decreasing by 3.34%. These further illustrate the key role of MGLS in enhancing network generalizability and feature dynamics perception.

**Analysis of Mask Guided Cross-Modal Cross-Attention:** After the removal of MCCA (i.e., AMG-Fuse w/o MCCA), the quantitative results are shown in Table 6, where all the metrics show a

significant decrease, with an average performance decrease of about 6.9%, which further validates the key role of MCCA in the proposed method.

Table 6: Ablation studies for the proposed components. The marked red indicates the best score.

| Source | Methods | $Q_{MI}$ | $Q_G$ | $Q_M$ | $VIF$ | $Q_{CB}$ | $SSIM$ | $Q^{AB/F}$ |
|---|---|---|---|---|---|---|---|---|
| Snow | w/o TDAS | 0.462 | 0.3951 | 0.5097 | 0.3268 | 0.4878 | 0.3961 | 0.5012 |
| | AMG-Fuse | 0.4960 | 0.4240 | 0.6408 | 0.3463 | 0.5062 | 0.4302 | 0.5519 |
| Rain | w/o MGLS | 0.3919 | 0.3910 | 0.6782 | 0.3531 | 0.4762 | 0.3928 | 0.4982 |
| | AMG-Fuse | 0.4084 | 0.4163 | 0.6940 | 0.3582 | 0.4879 | 0.4080 | 0.5184 |
| Haze | w/o MCCA | 0.2803 | 0.4178 | 0.5012 | 0.3184 | 0.4587 | 0.3616 | 0.5217 |
| | AMG-Fuse | 0.3096 | 0.4376 | 0.5438 | 0.3445 | 0.4864 | 0.4083 | 0.5414 |

## A.7 ADDITIONAL RESTORATION BASELINE COMPARISONS

Most existing multi-modality image fusion methods typically assume that the input images have been partially restored or that degradations have been removed. However, different restoration algorithms vary in their ability to remove rain, snow, haze, or noise, which may result in different levels of residual degradation in the inputs. To more comprehensively evaluate the robustness of fusion methods under complex weather conditions, we additionally introduce two restoration algorithms as pre-processing baselines. Specifically, DA-CLIP(Luo et al., 2024) is used for snow and rain scenes, while DehazeFormer(Song et al., 2023) is employed for haze scenarios. The outputs of these restoration algorithms are then used as inputs for fusion methods that have demonstrated strong performance in Table 1 and Table 2, namely EMMA, Text-IF, and SAGE, for comparative experiments.

The quantitative results are presented in the Table 7. It can be observed that although different restoration algorithms have some impact on fusion performance, the proposed method still maintains a leading performance. Moreover, this highlights a key limitation of the "restoration + fusion" pipeline: variations in restoration quality can directly affect subsequent fusion results, and residual degradations may be preserved or even amplified during fusion. In addition, it is often difficult to determine a single restoration method that is universally effective across different weather conditions. In contrast, the proposed unified framework can effectively handle multimodal information and robustly address complex degradations without relying on any specific restoration algorithm, making it a highly effective solution.

Table 7: Performance comparison of different fusion algorithms using multiple restoration baselines across snow, rain, and haze scenarios. The marked red indicates the best score and the marked green indicates the second score.

| Source | Methods | Pub. | Restoration | $Q_{MI}$ | $Q_G$ | $Q_M$ | $VIF$ | $Q_{CB}$ | $SSIM$ | $Q^{AB/F}$ |
|---|---|---|---|---|---|---|---|---|---|---|
| Snow | EMMA | CVPR 24 | DA-CLIP | 0.4301 | 0.2406 | 0.3501 | 0.1763 | 0.3882 | 0.2243 | 0.3361 |
| | Text-IF | CVPR 24 | | 0.4025 | 0.3508 | 0.4162 | 0.2549 | 0.3365 | 0.2906 | 0.4214 |
| | SAGE | CVPR 25 | | 0.4067 | 0.2216 | 0.3490 | 0.2946 | 0.3806 | 0.2286 | 0.2956 |
| | AMG-Fuse | – | × | 0.4960 | 0.4240 | 0.6408 | 0.3463 | 0.5062 | 0.4302 | 0.5519 |
| Rain | EMMA | CVPR 24 | DA-CLIP | 0.3664 | 0.2333 | 0.3918 | 0.2204 | 0.4355 | 0.2494 | 0.3172 |
| | Text-IF | CVPR 24 | | 0.3701 | 0.3771 | 0.4645 | 0.2784 | 0.4541 | 0.3591 | 0.4246 |
| | SAGE | CVPR 25 | | 0.3588 | 0.1772 | 0.3712 | 0.3024 | 0.3969 | 0.2058 | 0.2423 |
| | AMG-Fuse | – | × | 0.4084 | 0.4163 | 0.6940 | 0.3582 | 0.4879 | 0.4080 | 0.5184 |
| Haze | EMMA | CVPR 24 | DehazeFormer | 0.3518 | 0.4292 | 0.5010 | 0.3145 | 0.4719 | 0.3695 | 0.5408 |
| | Text-IF | CVPR 24 | | 0.3726 | 0.4422 | 0.4706 | 0.2895 | 0.4532 | 0.4062 | 0.5310 |
| | SAGE | CVPR 25 | | 0.3484 | 0.4007 | 0.5212 | 0.3421 | 0.4710 | 0.3718 | 0.5131 |
| | AMG-Fuse | – | × | 0.3096 | 0.4376 | 0.5438 | 0.3445 | 0.4864 | 0.4083 | 0.5414 |

## A.8 ADDITIONAL DOWNSTREAM TASK EVALUATION: SEMANTIC SEGMENTATION

Beyond evaluating different fusion methods on object detection to assess their ability to preserve salient target information, we also examine their capacity to retain global scene semantics through a semantic segmentation downstream task. Specifically, we employ BANet (Peng et al., 2021) on the MSRS dataset to compare the segmentation accuracy of images generated by various fusion

algorithms. As shown in the Table 8, our proposed method ranks within the top two across all six categories and achieves the highest overall mIoU. Although EMMA and Text-IF also show strong semantic retention capability, they struggle to maintain robustness under adverse weather degradations. In contrast, our method leverages a mask-guided learning strategy that effectively strengthens multi-modal feature interaction, enabling the extraction of more discriminative semantic information and substantially enhancing the semantic expressiveness of the fused results.

Table 8: Segmentation accuracy comparison on the MSRS dataset.

| Methods | Pub. | Background | Car | Person | Bike | Curve | Car Stop | Guardrail | Color cone | Bump | mIoU |
|---|---|---|---|---|---|---|---|---|---|---|---|
| LRRNet | *PAMI 23* | 98.34 | 89.09 | 68.11 | 69.29 | 52.04 | 71.57 | 81.97 | 64.28 | 78.27 | 74.77 |
| Text-DiFuse | *NIPS 24* | 98.45 | 88.96 | 71.04 | 71.94 | 60.05 | 72.23 | 84.03 | 65 | 78.59 | 76.7 |
| EMMA | *CVPR 24* | 98.53 | 90.36 | 74.65 | 71.61 | 64.61 | 74.49 | 83.73 | 65.61 | 76.36 | 77.77 |
| Text-IF | *CVPR 24* | 98.51 | 90.27 | 73.01 | 72.17 | 64.86 | 74.11 | 84.88 | 66.24 | 78.98 | 78.12 |
| GIFNet | *CVPR 25* | 98.4 | 88.98 | 71.13 | 71.92 | 59.19 | 74.11 | 78.05 | 65.3 | 64.62 | 74.63 |
| SAGE | *CVPR 25* | 98.38 | 88.8 | 69.53 | 70.37 | 57.13 | 71.21 | 86.06 | 65.08 | 76.9 | 75.94 |
| AWFusion | *Arxiv 24* | 98.52 | 89.5 | 71.8 | 71.75 | 64.17 | 74.18 | 83.67 | 66.01 | 77.53 | 77.46 |
| AMG-Fuse | - | 98.54 | 89.69 | 72.63 | 72.26 | 63.86 | 74.13 | 85.89 | 66.27 | 80.31 | 78.18 |

## A.9 Mask Learning on a Tiny Variant: Efficiency and Generalization

In this study, we validate that the proposed mask learning strategy can significantly improve the learning efficiency of the network. It enables the model to better capture the dynamic interaction mechanisms between multimodal features while effectively suppressing overfitting risks.

For lightweight models, whose parameter capacity is limited, the amount and diversity of perceptible features are substantially reduced compared with the original large-scale model. As a result, these models are more prone to learning static feature distributions, which may lead to overfitting or difficulties in model fitting. Under such circumstances, if the proposed mask learning strategy is truly effective, its advantages should become even more pronounced when applied to lightweight architectures.

Motivated by this consideration, we introduce a lightweight structural redesign of the original model and construct a Tiny variant. Our test images have a resolution of $224 \times 224$, and the detailed descriptions are as follows:

**Tiny Model Variant:**

- The initial channel dimension processed by the HTB is reduced from 48 to 16.
- The number of Transformer blocks is changed from [4, 6, 6, 8] to [1, 2, 2, 4].
- The number of attention heads is modified from [2, 4, 6, 8] to [1, 2, 2, 4].

The resulting model contains 4.91M parameters with a total of 20.972 GFLOPs.

We conducted a comprehensive evaluation of the lightweight model under three typical adverse weather scenarios, and the results are presented in Table 9. Since the main paper already provides a full quantitative comparison against all competing methods, the appendix reports only the average ranking of the lightweight AMG-Tiny model and all comparative methods across three adverse weather conditions.

It can be observed that AMG-Tiny still achieves top-ranked performance under adverse weather conditions. Although its overall scores are slightly lower than those of Text-IF, it is worth noting that Text-IF has 249.707 GFLOPs and 89.014M parameters, whereas the proposed AMG-Tiny achieves reductions of approximately 11.9× in FLOPs and 18.1× in parameters, demonstrating a substantial efficiency advantage. This indicates that the mask-guided learning strategy can enhance model robustness under degradations even with substantially reduced computation and parameters.

**Efficiency and Inference Performance:** Notably, AMG-Tiny achieves significant gains in both structural and inference efficiency while maintaining competitive performance:

- Original model: 242.03 GFLOPs, 59.74M parameters
- Tiny variant: 20.972 GFLOPs, 4.91M parameters
- Reductions: 11.54× (FLOPs) and 12.17× (parameters)

During inference, the model processes a single $224 \times 224$ image in 40.302 ms on average (standard deviation 0.480 ms), corresponding to approximately 24.81 FPS, demonstrating near real-time capability despite its lightweight design. These results further indicate that the mask-guided learning mechanism not only improves feature allocation but also substantially enhances the model's scalability and practicality in resource-constrained environments.

Table 9: Quantitative performance of the lightweight AMG-Tiny model across three adverse weather conditions. The values represent the metric scores, and the numbers in parentheses indicate the ranking of the Tiny model when compared against all comparative methods reported in the main paper.

| Methods | Pub. | Restor. | $Q_{MI}$ | $Q_G$ | $Q_M$ | $VIF$ | $Q_{CB}$ | $SSIM$ | $Q^{AB/F}$ |
|---------|------|---------|----------|-------|-------|-------|----------|--------|-----------|
| LRRNet | PAMI 23 | | 0.3606 | 0.2733 | 0.3810 | 0.2457 | 0.3767 | 0.1350 | 0.3668 |
| Text-DiFuse | NIPS 24 | | 0.3312 | 0.2875 | 0.3581 | 0.2329 | 0.4188 | 0.2606 | 0.3827 |
| EMMA | CVPR 24 | AdaIR | 0.4072 | 0.3844 | 0.5125 | 0.3065 | 0.4788 | 0.3697 | 0.4984 |
| Text-IF | CVPR 24 | | 0.3476 | 0.4094 | 0.5578 | 0.3189 | 0.4800 | 0.3929 | 0.5150 |
| GIFNet | CVPR 25 | | 0.2908 | 0.3127 | 0.3391 | 0.2741 | 0.4824 | 0.3553 | 0.3634 |
| SAGE | CVPR 25 | | 0.3667 | 0.2950 | 0.4595 | 0.3382 | 0.4312 | 0.3175 | 0.4197 |
| AWFusion | Arxiv 24 | x | 0.3605 | 0.3723 | 0.5811 | 0.3152 | 0.4700 | 0.3721 | 0.4959 |
| AMG-Tiny | – | | 0.3713 | 0.3851 | 0.5422 (3) | 0.3327 | 0.4776 (4) | 0.3887 | 0.4965 (3) |

## A.10 VISUALIZATION OF NETWORK OUTPUTS

Figure 9 shows the three outputs of the network in three types of adverse weather: visible features, infrared features and fusion results, with the corresponding "Pseudo Ground Truth" provided as a reference for comparison.

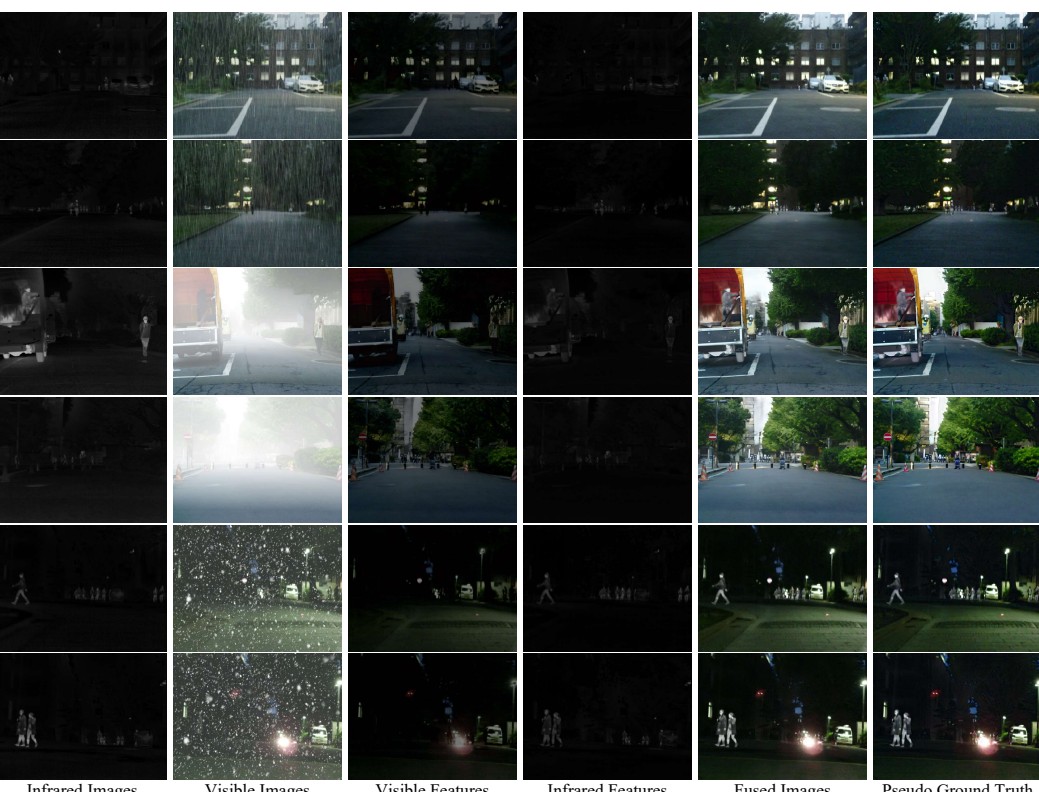

| Infrared Images | Visible Images | Visible Features | Infrared Features | Fused Images | Pseudo Ground Truth |

Figure 9: Visualization of Network Outputs and "Pseudo Ground Truth".

## A.11 LIMITATIONS AND FUTURE WORK

In this work, we utilize a Mask to guide the proposed algorithm to learn the implicit modal allocation knowledge in the "Pseudo Ground Truth". In addition, we propose the Task-Coupled Degradation-Aware Learning Strategy (TDAS) to constrain the network to reconstruct clear details, which proves to be both simple and effective under adverse weather conditions. However, our experiments reveal that for certain types of degradation (e.g., noise), the mask fails to accurately capture the degradation distribution. As shown in Figure 10, neither $Mask \times VI$ nor $(1 - Mask) \times IR$ can effectively remove noise from the input. In future work, we will further refine our method to improve its generalization to more complex degradation scenarios.

Moreover, our experiments reveal that the use of the Histogram Transformer in the Mask-Guided Feature Extraction Module introduces the high computational complexity characteristic of Transformer architectures into our method. Specifically, for an input size of $224 \times 224$, the proposed method incurs $242.03G$ FLOPs and $59.74M$ parameters. In future work, we plan to further optimize the computational efficiency of our model to enhance its applicability in real-world scenarios.

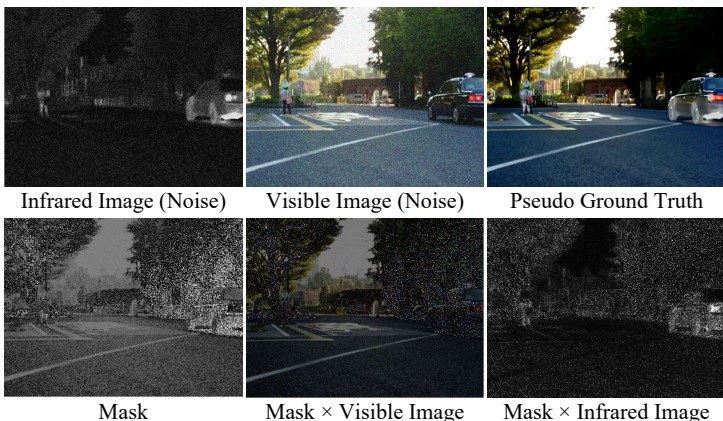

| Infrared Image (Noise) | Visible Image (Noise) | Pseudo Ground Truth |
| Mask | Mask × Visible Image | Mask × Infrared Image |

Figure 10: Decomposition Experiment in Noise Scene.

## A.12 QUALITATIVE COMPARISON RESULTS IN STANDARD DATASETS

As shown in Figure 11 and Figure 12, the proposed algorithm achieves excellent quantitative performance in the LLVIP and M3FD datasets, verifying its robustness in clean scenarios.

## A.13 MORE QUALITATIVE COMPARISON RESULTS IN AWMM-100K DATASET

As shown in Figure 13, Figure 14 and Figure 15, we additionally present more fusion results of different algorithms under adverse weather conditions.

## A.14 BROADER IMPACTS

Multi-modality image fusion is a core task in low-level vision, which is of great value for improving perception in complex environments. In this paper, we propose a novel image fusion framework that achieves more stable and high-quality fusion performance under adverse weather conditions. The method has a wide potential for practical applications, especially in areas with high requirements for environment perception, such as autonomous driving, which helps to provide more accurate, comprehensive, and robust information support.

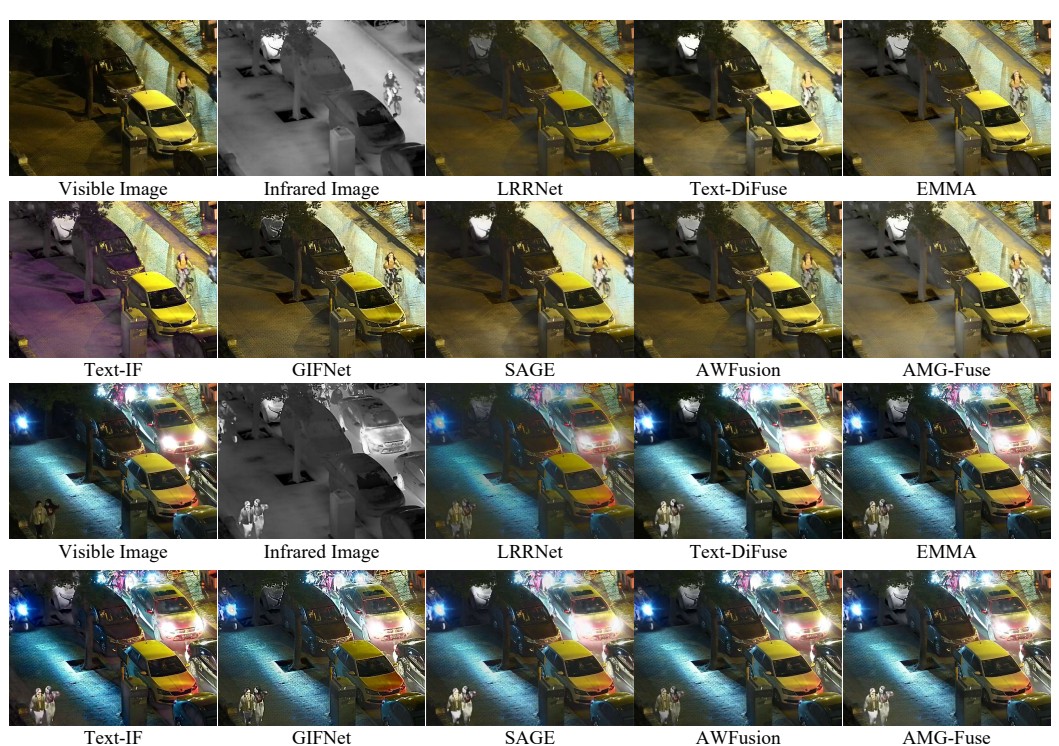

Figure 11: Qualitative comparison results of all methods in LLVIP dataset

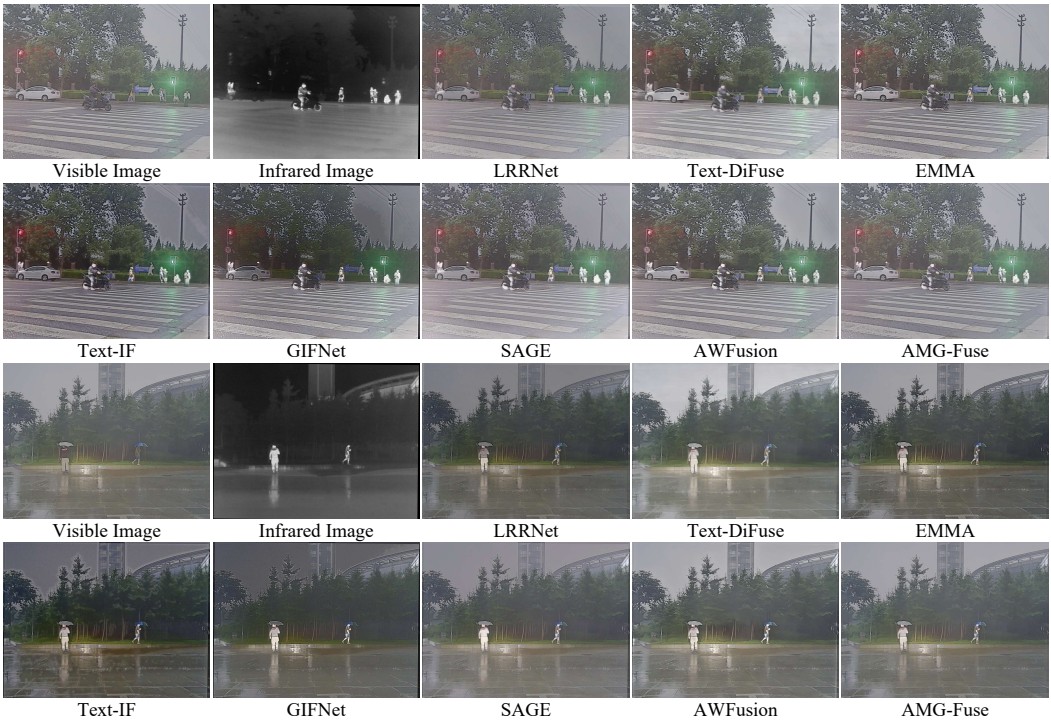

Figure 12: Qualitative comparison results of all methods in M3FD dataset.

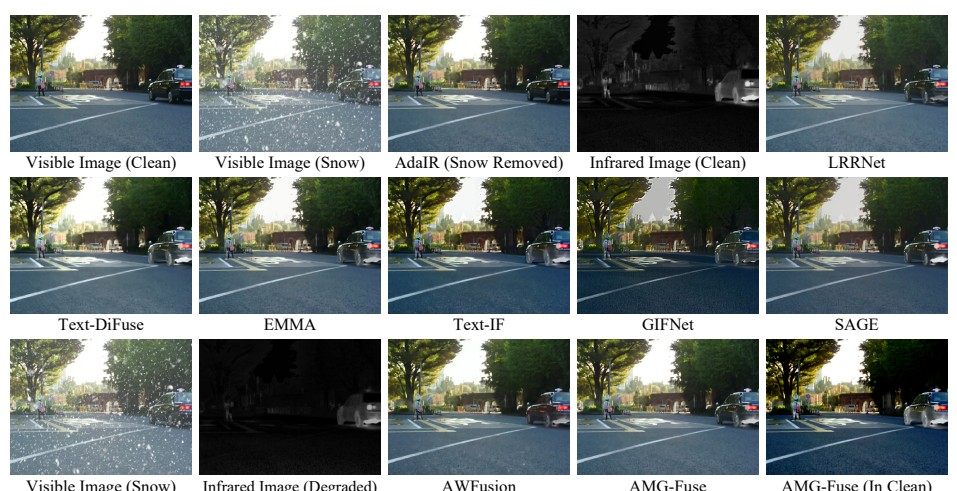

Figure 13: Qualitative comparison results of all methods in snow weather.

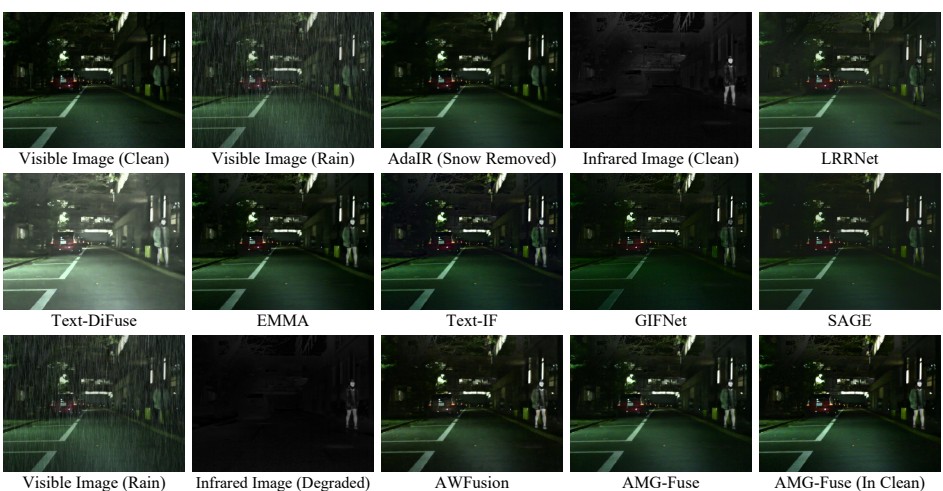

Figure 14: Qualitative comparison results of all methods in rain weather.

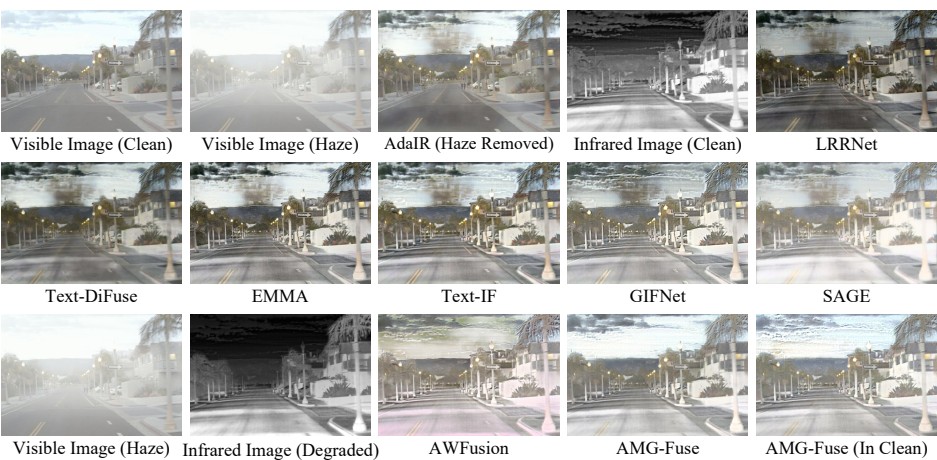

Figure 15: Qualitative comparison results of all methods in haze weather.

