# OpenReview forum: "Multi-modality Image Fusion under Adverse Weather: Mask-Guided Feature Restoration and Interaction"
_ICLR.cc/2026/Conference — Submitted to ICLR 2026_

### Official Review · Reviewer_AjCT · 2025-10-30

**Soundness:** 3
**Presentation:** 3
**Contribution:** 2
**Rating:** 4
**Confidence:** 4

**Summary:**

This paper introduces a novel multi-modality image fusion method designed to address image fusion challenges under adverse weather conditions. The proposed method integrates feature restoration with cross-modal interaction through a mask-guided mechanism, utilizing "Pseudo Ground Truth" to simplify the training process. This helps in recovering degraded features and enhancing the interaction between multi-modal features. Key innovations include the construction of a mask that quantifies each modality's contribution during the fusion process, enabling effective separation and interaction of features. Additionally, the method introduces a Mask-Guided Feature Extraction Module (MFEM), Mask-Guided Learning Strategy (MGLS), and Task-Coupled Degradation-Aware Learning Strategy (TDAS) to improve feature restoration and cross-modal interaction. Extensive experiments on both synthetic and real-world datasets under adverse weather conditions demonstrate that AMG-Fuse outperforms state-of-the-art methods in visual quality, quantitative metrics, and downstream tasks, while also showing strong adaptability in ideal conditions.

**Strengths:**

1.The AMG-Fuse integrates feature restoration and cross-modal interaction into a unified framework to tackle the challenges of image fusion under adverse weather conditions. The mask-guided mechanism combined with "Pseudo Ground Truth" allows the network to learn dynamic cross-modal information allocation, overcoming the limitations of traditional methods relying on static modality distribution.

2.The proposed method is thoroughly tested on both synthetic and real-world datasets under various weather conditions. The results demonstrate AMG-Fuse’s superiority in visual quality, quantitative metrics, and downstream tasks, proving its robustness and applicability across different environments.

3.While AMG-Fuse excels in handling adverse weather, it also performs well in ideal environments, showcasing its generalization ability and versatility in a variety of conditions.

**Weaknesses:**

1.The inclusion of the Histogram Transformer module and depthwise separable convolutions leads to high computational demands and a large number of parameters, which may limit its applicability in real-time scenarios with resource constraints.

2.The overall architecture of this article is similar to a simple stacking of modules, which also includes a frozen restoration model in the middle. The understanding of the essential relationship and differences between restoration and fusion tasks is not yet thorough enough.

3.While the method performs well in adverse weather conditions, it struggles with noise and other degradation types, as the mask fails to accurately capture the degradation distribution in such cases, leading to suboptimal performance.

4.The degradation targeted in this article mainly includes rain, snow, and haze, and the rain and snow part mainly uses artificial virtual data, which is not realistic enough in terms of visual effects. Therefore, it is difficult to reflect the value in practical applications. The method may require further optimization for handling other types of degradation, such as motion blur or strong lighting variations.

5.The method involves multiple strategies that require careful tuning of hyperparameters, which increases the complexity of the training process and experimental design.

**Questions:**

1.As shown in the Haze section of Fig.6, the restoration results of AdaIR are obviously not excellent enough. Can the author choose other restoration models for comparative experiments and prove that the proposed method is indeed more effective than the method of two models.

2.Some of the results in the ablation experiment in this article do not reflect the significance of this module, especially in Mask Guided Learning Strategy, where the contours of the characters are very blurry. The author needs to submit other figures to prove it.

3.In the author's comparison method, only Text-if and AWFusion are integrated restoration fusion methods. There have been many works in this area in the past year, and the author should provide more reference methods.

4.The author needs to provide performance on real-world datasets to validate the practical value of the method.

5.The author did not provide visual results for object detection, and relevant content needs to be supplemented.

6.Is it possible to use model compression or quantization techniques to reduce the computational overhead during inference and improve the feasibility of this method in real-time applications for high computational complexity problems?

7.If the degree of degradation changes, will the model's processing capability weaken and will the area of focus of Mask change?

---

> ### Author Response · Authors · 2025-11-20
> **Response to Weakness 1, Question 6, and Weakness 2 (About model complexity, architectural design, and the relationship between restoration and fusion)**
>
> *We sincerely thank Reviewer AjCT for the time and effort dedicated to evaluating our work. Your comments primarily address model efficiency, algorithm design, and comparative experiments. Throughout the process of responding to your suggestions, our manuscript has been substantially improved, and we greatly appreciate your professional guidance. Below, we provide point-by-point responses to your comments. If our clarifications adequately address your concerns, we hope you may consider raising your evaluation score.*
>
> **The revisions have been incorporated into the manuscript, and the latest version is now available for your review.**
>
> `W1 The inclusion of the Histogram Transformer module and depthwise separable convolutions leads to high computational demands and a large number of parameters, which may limit its applicability in real-time scenarios with resource constraints.`
>
> `Q6 Is it possible to use model compression or quantization techniques to reduce the computational overhead during inference and improve the feasibility of this method in real-time applications for high computational complexity problems?`
>
> We thank the reviewer for raising concerns regarding the computational efficiency of our method. Since our primary objective is to achieve high-quality multi-modal fusion under complex and adverse conditions. Therefore, the work was not initially oriented toward low-power deployment scenarios. Nonetheless, we have provided a lightweight model design and corresponding performance comparisons, which can be found in our responses to **Reviewer cVus (W2 and W3)**.
>
>
> `W2. The overall architecture of this article is similar to a simple stacking of modules, which also includes a frozen restoration model in the middle. The understanding of the essential relationship and differences between restoration and fusion tasks is not yet thorough enough.`
>
>
> We thank the reviewer for their attention to our architecture. Our design is not a simple stacking of modules; rather, the entire framework is built around the role of Mask, which guides the organization of restoration and fusion. We provide the following clarification to address potential misunderstandings.
>
> 1.**Overall Architecture Design**
>
> Our network is not a simple stack of modules; it is carefully designed based on our observations of the role of masks in modality feature allocation and degradation awareness. For instance, the visualizations in Figures 4 and 5 show that masks effectively serve as guidance for modality feature allocation and interaction while also capturing degraded regions. These results directly motivate the design of our MFEM and MGLS modules.
>
> 2.**Frozen Restoration Model**
>
> The frozen restoration model is used solely as an auxiliary supervisory signal (loss function) to encourage the fusion network to produce clear, degradation-free outputs. It does not directly interact with the feature flow in the fusion process. The key idea is that, for undegraded input images, the restoration network tends to generate outputs highly consistent with the original images. We leverage this property by comparing pre- and post-restoration outputs as a supervisory signal to guide the fusion network in learning more robust clear features, rather than using it as part of the fusion process itself.
>
> 3.**Relationship and Distinction Between Restoration and Fusion**
>
> We have carefully studied the intrinsic relationship and differences between these two tasks. Multi-modal image fusion aims to extract complementary information from different modalities, remove redundancy, and reconstruct a discriminative composite representation. In contrast, image restoration focuses on recovering fine details and textures within a single modality, aiming for pixel-level fidelity. The optimization objectives of the two tasks are therefore fundamentally different. In a unified restoration-fusion framework, pursuing full pixel-level restoration is not necessary. Accordingly, we primarily leverage the mask to guide the network in learning modality feature allocation, and we design TDAS to enable the network to learn clear features in a more sophisticated manner, achieving a balanced and synergistic integration of restoration and fusion objectives.

---

> ### Author Response · Authors · 2025-11-20
> **Response to Weakness 3 and Weakness 5 (About noise-related limitation and hyperparameter)**
>
> `W3. While the method performs well in adverse weather conditions, it struggles with noise and other degradation types, as the mask fails to accurately capture the degradation distribution in such cases, leading to suboptimal performance.`
>
> We thank the reviewer for highlighting the limitations of the proposed method. As noted in **Appendix A.11: Limitations and Future Work**, the mask learning strategy may experience performance degradation in noise-dominated scenarios. This is because the strategy primarily relies on masks to perceive and allocate degraded regions, while random noise typically lacks stable spatial distribution patterns, making it difficult for the mask to accurately model the degradation. In future work, we plan to explore explicit modeling mechanisms for unstructured noise to further enhance the robustness of the algorithm.
>
> It is important to emphasize that this phenomenon reflects a limitation of the method rather than a flaw in the algorithm itself. If the mask learning strategy is removed and the network is trained directly using clean images as strong supervision, it can still generate noise-free fusion results. However, this approach diminishes the network’s ability to learn dynamic modality feature allocation and interaction, which contradicts the original design goal of adaptive multi-modal fusion.
>
> Overall, we believe that different learning strategies have their respective strengths and applicable scenarios. The proposed mask-guided strategy excels in scenarios with structured degradations, such as adverse weather conditions, while there remains room for improvement under noise-dominated complex degradations. Addressing this challenge will be a focus of our future research.
>
> `W5. The method involves multiple strategies that require careful tuning of hyperparameters, which increases the complexity of the training process and experimental design.`
>
> We thank the reviewer for the attention to hyperparameter settings. We clarify that the proposed method does not rely on complex hyperparameter tuning:
>
> 1. **Network architecture**: The algorithm uses a Transformer for feature extraction, consistent with mainstream Transformer-based methods. Core parameters, *dim*, *num_blocks*, *num_refinement_blocks*, and *heads*, are standard Transformer configurations, not specially designed, and thus do not increase model complexity.
> `
> 2. **Learning strategy**: MGLS introduces only a decay coefficient that updates adaptively during training without manual tuning. TDAS contains no manually adjusted hyperparameters.
>
> 3. **Loss function**: All loss weights are set to 1, with no manual weighting or fine-tuning, keeping training simple and stable.
>
> In summary, the proposed method follows a concise and efficient design, with minimal hyperparameters that are standard and do not add training complexity.

---

> ### Author Response · Authors · 2025-11-20
> **Response to Weakness 4 and Question 4 (About realism of synthetic degradations and evaluations on real data)**
>
> `W4. The degradation targeted in this article mainly includes rain, snow, and haze, and the rain and snow part mainly uses artificial virtual data, which is not realistic enough in terms of visual effects. Therefore, it is difficult to reflect the value in practical applications. The method may require further optimization for handling other types of degradation, such as motion blur or strong lighting variations.`
>
> `Q4. The author needs to provide performance on real-world datasets to validate the practical value of the method.`
>
>
> We thank the reviewer for valuable comments regarding the practical applicability of our method. We agree with the concern about the lack of real paired rainy and snowy datasets. We would like to emphasize that, at present, there is still a scarcity of large-scale, well-registered infrared–visible datasets containing real rain and snow, which to some extent limits comprehensive evaluation of models in real-world scenarios.
>
> Regarding data realism, we note that the synthetic rain and snow images in the AWMM-100K dataset are generated based on masks extracted from real rain and snow video sequences. These masks are then used in a rendering process to synthesize degraded images, a strategy widely adopted in the deraining literature [*Towards unified deep image deraining: A survey and a new benchmark, TPAMI 2025.*]. This synthesis approach preserves the structural characteristics of real rain and snow, and its statistical and visual properties are reasonably close to real degradations. Therefore, training on these datasets allows the model to generalize effectively to real-world scenarios without compromising practical utility.
> In addition, to further demonstrate the real-world applicability of the proposed algorithm, we conducted supplementary experiments on the real-world data within AWMM-100K, which captures authentic haze effects. These experiments are reported in Section 4.3 of the main text, as detailed below:
>
> ---
>
> > ### Additional Experiments on Real-World Data
> >
> > To further validate the applicability of the proposed algorithm on real-world data, we conducted experiments on the AWMM-100k real haze dataset, with quantitative results shown in Table 2. The proposed method achieves top-two scores across all metrics, demonstrating the effectiveness of the mask-guided learning strategy. This strategy prevents the model from merely learning static feature recovery, enabling it to restore fine details while facilitating multi-modality information interaction.
> >
> > **Table2: Quantitative comparison of all methods in real-world scenarios.**
> >
> > | Methods     | Pub.            | Q_MI   | Q_G     | Q_M    | VIF     | Q_CB    | SSIM   | Q^AB/F  |
> > |------------|-----------------|--------|---------|--------|---------|---------|--------|---------|
> > | LRRNet     | *PAMI 23*       | 0.5429 | 0.3944  | 0.9170 | 0.4079  | 0.4694  | 0.4174 | 0.4773  |
> > | Text-DiFuse| *NIPS 24*       | 0.5174 | 0.3325  | 0.9384 | 0.3850  | 0.4108  | 0.3848 | 0.4657  |
> > | EMMA       | *CVPR 24*       | 0.5497 | 0.3884  | 0.8056 | 0.3264  | 0.4470  | 0.4342 | 0.5301  |
> > | Text-IF    | *CVPR 24*       | 0.5258 | 0.5748  | 1.0852 | 0.3566  | 0.4936  | 0.4972 | 0.6800  |
> > | GIFNet     | *CVPR 25*       | 0.5085 | 0.3673  | 0.8833 | 0.3021  | 0.4250  | 0.4489 | 0.4605  |
> > | SAGE       | *CVPR 25*       | 0.5200 | 0.4786  | 1.0538 | 0.3647  | 0.4298  | 0.4816 | 0.5314  |
> > | AWFusion   | *Arxiv 24*      | 0.5032 | 0.4547  | 1.0028 | 0.3523  | 0.4649  | 0.5036 | 0.5860  |
> > | AMG-Fuse   | \_              | **0.7234** | 0.5114 (2)  | **1.1022** | **0.4373**  | 0.4786 (2)  | **0.5202** | 0.5957 (2)  |
>
> ---
> **The complete content can be found in Section 4.3.**
>
> Additionally, due to the lack of publicly available large-scale datasets for motion blur or strong illumination changes, we could not conduct targeted experiments for such degradations. However, the MSRS dataset contains images with these real-world effects. For instance, motion blur appears in images: 00001D, 00009N, 00256D, 00445D, 00449D, 00458D, 00484D; strong illumination variations appear in images: 00037N, 00822N, 00909N, 01185N, 01188N, 01196N, 01203N. Table 5 reports a comparison of different algorithms on MSRS dataset, demonstrating the practical value of the proposed method.
>
> ---
>
> We hope these experiments address the reviewers' concerns regarding the real-world applicability of our algorithm.

---

> ### Author Response · Authors · 2025-11-20
> **Response to Question 1 (About additional restoration baselines)**
>
> `Q1. As shown in the Haze section of Fig.6, the restoration results of AdaIR are obviously not excellent enough. Can the author choose other restoration models for comparative experiments and prove that the proposed method is indeed more effective than the method of two models.`
>
>
> We thank the reviewer for the question. To further enhance our comparison experiments, we added DA-CLIP and DehazeFormer as restoration models. This content has been added to **Appendix A.7**, as follows:
>
> > ### Appendix A.7: Additional Restoration Baseline Comparisons
>
> > Most existing multi-modality image fusion methods typically assume that the input images have been partially restored or that degradations have been removed. However, different restoration algorithms vary in their ability to remove rain, snow, haze, or noise, which may result in different levels of residual degradation in the inputs. To more comprehensively evaluate the robustness of fusion methods under complex weather conditions, we additionally introduce two restoration algorithms as pre-processing baselines. Specifically, DA-CLIP is used for snow and rain scenes, while DehazeFormer is employed for haze scenarios. The outputs of these restoration algorithms are then used as inputs for fusion methods that have demonstrated strong performance in Table 1 and Table 2, namely EMMA, Text-IF, and SAGE, for comparative experiments.
>
> > The quantitative results are presented in the Table 7. It can be observed that although different restoration algorithms have some impact on fusion performance, the proposed method still maintains a leading performance. Moreover, this highlights a key limitation of the “restoration + fusion” pipeline: variations in restoration quality can directly affect subsequent fusion results, and residual degradations may be preserved or even amplified during fusion. In addition, it is often difficult to determine a single restoration method that is universally effective across different weather conditions. In contrast, the proposed unified framework can effectively handle multimodal information and robustly address complex degradations without relying on any specific restoration algorithm, making it a highly effective solution.
>
>
> > **Tab. Performance comparison of different fusion algorithms using multiple restoration baselines across snow, rain, and haze scenarios.**
> >
> > ---
> >
> > | Source | Methods   | Pub.         | Restoration     | Q_MI        | Q_G        | Q_M        | VIF        | Q_CB       | SSIM       | Q^AB/F     |
> > |--------|-----------|-------------|----------------|------------|------------|------------|------------|------------|------------|------------|
> > | Snow   | EMMA      | CVPR 24     | DA-CLIP        | *0.4301*   | 0.2406     | 0.3501     | 0.1763     | *0.3882*   | 0.2243     | 0.3361     |
> > | Snow   | Text-IF   | CVPR 24     | DA-CLIP        | 0.4025     | *0.3508*   | *0.4162*   | 0.2549     | 0.3365     | *0.2906*   | *0.4214*   |
> > | Snow   | SAGE      | CVPR 25     | DA-CLIP        | 0.4067     | 0.2216     | 0.3490     | *0.2946*   | 0.3806     | 0.2286     | 0.2956     |
> > | Snow   | AMG-Fuse  | _           | ×              | **0.4960** | **0.4240** | **0.6408** | **0.3463** | **0.5062** | **0.4302** | **0.5519** |
> > | Rain   | EMMA      | CVPR 24     | DA-CLIP        | 0.3664     | 0.2333     | 0.3918     | 0.2204     | 0.4355     | 0.2494     | 0.3172     |
> > | Rain   | Text-IF   | CVPR 24     | DA-CLIP        | *0.3701*   | *0.3771*   | *0.4645*   | 0.2784     | *0.4541*   | *0.3591*   | *0.4246*   |
> > | Rain   | SAGE      | CVPR 25     | DA-CLIP        | 0.3588     | 0.1772     | 0.3712     | *0.3024*   | 0.3969     | 0.2058     | 0.2423     |
> > | Rain   | AMG-Fuse  | _           | ×              | **0.4084** | **0.4163** | **0.6940** | **0.3582** | **0.4879** | **0.4080** | **0.5184** |
> > | Haze   | EMMA      | CVPR 24     | DehazeFormer   | *0.3518*   | 0.4292     | 0.5010     | 0.3145     | *0.4719*   | 0.3695     | *0.5408*   |
> > | Haze   | Text-IF   | CVPR 24     | DehazeFormer   | **0.3726** | **0.4422** | 0.4706     | 0.2895     | 0.4532     | *0.4062*   | 0.5310     |
> > | Haze   | SAGE      | CVPR 25     | DehazeFormer   | 0.3484     | 0.4007     | *0.5212*   | *0.3421*   | 0.4710     | 0.3718     | 0.5131     |
> > | Haze   | AMG-Fuse  | _           | ×              | 0.3096     | *0.4376*   | **0.5438** | **0.3445** | **0.4864** | **0.4083** | **0.5414** |
>
> **The complete content can be found in Appendix A.7.**

---

> ### Author Response · Authors · 2025-11-20
> **Response to Question 2 and Question 3 (About clearer ablation evidence and recent restoration–fusion baselines)**
>
> `Q2. Some of the results in the ablation experiment in this article do not reflect the significance of this module, especially in Mask Guided Learning Strategy, where the contours of the characters are very blurry. The author needs to submit other figures to prove it.`
>
> We thank the reviewer for the attention and feedback on the ablation study. As shown in Figure 7, removing the MGLS module leads to noticeable loss of details and decreased contrast in the fused results. The blurring issue noted by the reviewer mainly originates from the low resolution of the source images, rather than the proposed method.
>
> Moreover, quantitative analysis in Table 6 of the Appendix further supports this. When MGLS is removed, although human-vision-based metrics (e.g., VIF and Q_CB) show limited degradation due to the presence of "pseudo-ground truth," metrics more sensitive to structural features (e.g., Q_G and Q_M) degrade significantly, with an average drop of 3.34% across all metrics. These results further confirm the critical role of MGLS in enhancing network generalization and feature dynamic awareness.
>
> In summary, we hope the reviewer will reconsider their evaluation of the ablation study.
>
> `Q3. In the author's comparison method, only Text-if and AWFusion are integrated restoration fusion methods. There have been many works in this area in the past year, and the author should provide more reference methods.`
>
> We thank the reviewer for the attention to the comparative methods in our experiments. To further strengthen our study, we have included two additional image fusion algorithms for complex scenarios, UMCFuse (TIP 2025) and OmniFuse (TPAMI 2025). The results are summarized in the table below, with the highest scores in each row **bolded**:
>
>
> | Source | Methods   | Pub.     | Restoration | Q_MI       | Q_G        | Q_M        | VIF       | Q_CB       | SSIM      | Q^AB/F     |
> |--------|-----------|----------|------------|-----------|-----------|-----------|-----------|-----------|-----------|-----------|
> | Snow   | UMCFuse   | TIP 25   | AdaIR      | 0.3957    | 0.4109    | 0.4239    | 0.3127    | 0.4891    | 0.4168    | 0.4427    |
> |  Snow         | OmniFuse  | PAMI 25  |    AdaIR        | 0.4041    | 0.2729    | 0.3845    | 0.2340    | 0.4114    | 0.2515    | 0.3977    |
> |  Snow         | AMG-Fuse  | _        | x          | **0.4960**| **0.4240**| **0.6408**| **0.3463**| **0.5062**| **0.4302**| **0.5519**|
> | Rain   | UMCFuse   | TIP 25   | AdaIR      | 0.3678    | 0.3500    | 0.4626    | 0.3235    | 0.4562    | 0.3681    | 0.3954    |
> |   Rain        | OmniFuse  | PAMI 25  |  AdaIR          | 0.3642    | 0.2191    | 0.4027    | 0.2197    | 0.3774    | 0.1771    | 0.3340    |
> |  Rain         | AMG-Fuse  | _        | x          | **0.4084**| **0.4163**| **0.6940**| **0.3582**| **0.4879**| **0.4080**| **0.5184**|
> | Haze   | UMCFuse   | TIP 25   | AdaIR      | 0.1951    | 0.3490    | 0.3521    | 0.2580    | 0.4553    | 0.3481    | 0.3859    |
> |   Haze        | OmniFuse  | PAMI 25  |   AdaIR         | 0.2247    | 0.2638    | 0.3448    | 0.2204    | 0.4226    | 0.2558    | 0.3446    |
> |  Haze         | AMG-Fuse  | _        | x          | **0.3096**| **0.4376**| **0.5438**| **0.3445**| **0.4864**| **0.4083**| **0.5414**|
> | MSRS   | UMCFuse   | TIP 25   | AdaIR      | 0.4415    | 0.5235    | 0.6683    | 0.3621    | 0.5266    | 0.4851    | 0.5304    |
> |  MSRS         | OmniFuse  | PAMI 25  |  AdaIR          | 0.4483    | 0.2960    | 0.4117    | 0.2799    | 0.4520    | 0.2910    | 0.4243    |
> |  MSRS        | AMG-Fuse  | _        | x          | **0.7594**| **0.5952**| **1.5051**| **0.4706**| **0.5809**| **0.5044**| **0.6749**|
> | M3FD   | UMCFuse   | TIP 25   | AdaIR      | 0.4847    | 0.5104    | 0.7247    | 0.3500    | **0.5276**    | 0.4998    | 0.5569    |
> |  M3FD         | OmniFuse  | PAMI 25  |    AdaIR        | 0.4947    | 0.2655    | 0.4373    | 0.3510    | 0.4760    | 0.2990    | 0.4145    |
> |   M3FD        | AMG-Fuse  | _        | x          | **0.7010**| **0.5376**| **1.2994**| **0.4713**|  0.4979       | **0.5069**| **0.6384**|
> | LLVIP  | UMCFuse   | TIP 25   | AdaIR      | 0.3666    | 0.5940    | 0.5305    | 0.3209    | **0.4821**    | **0.4719**    | 0.6253    |
> |   LLVIP       | OmniFuse  | PAMI 25  |    AdaIR        | 0.3784    | 0.2658    | 0.1395    | 0.2360    | 0.4279    | 0.2611    | 0.3258    |
> |    LLVIP      | AMG-Fuse  | _        | x          | **0.4684**| **0.6228**| **0.7938**| **0.3952**|   0.4479    | 0.4574| **0.6907**|

---

> ### Author Response · Authors · 2025-11-20
> **Response to Question 5 and 7 (About object-detection results and model in different degradation levels)**
>
> `Q5. The author did not provide visual results for object detection, and relevant content needs to be supplemented.`
>
> We are grateful to the reviewer for the attention to the visualisation results of object detection. In response to this point, we have supplemented the visual results of object detection **in Figure 8 of the main text**.
>
> `Q7. If the degree of degradation changes, will the model's processing capability weaken and will the area of focus of Mask change?`
>
> We thank the reviewer for the attention to the model’s performance under varying degradation levels. First, regarding the model’s ability to handle different degradation intensities, this has been fully considered during training. Specifically, the AWMM-100k dataset is categorized into three levels, Light, Medium, and Heavy, based on degradation severity. We train the model using mixed-degradation samples to enhance its adaptability across different levels, enabling it to maintain stable fusion performance under varying degradation conditions.
>
> Second, concerning the variation of Mask focus regions, **as shown in Figure 5**, the Mask dynamically adjusts according to the distribution and density of degraded areas. As described in the paper, the Mask not only guides cross-modal feature allocation but also effectively perceives and locates degraded regions, thereby improving the model’s adaptability to complex degradation.

---

### Official Review · Reviewer_Gdro · 2025-10-31

**Soundness:** 3
**Presentation:** 4
**Contribution:** 3
**Rating:** 6
**Confidence:** 5

**Summary:**

Authors proposed a mask-guided multi-modal image fusion method designed to handle degradations under adverse weather. The main contributions are the proposed mask-guided feature extraction module and task-coupled degradation-aware learning strategy, which alleviate the optimization difficulty of jointly learning cross-modal interactions and feature restoration in complex scenes. Experiments show that the proposed method consistently outperforms the state-of-the-art methods on both image fusion and downstream tasks.

**Strengths:**

(1) By explicitly separating modality-specific components within the fused image, the network dynamically learns how to allocate multimodal information. This differs from existing mask-based fusion methods and is both novel and effective.

(2) The proposed mask-guided feature extraction module leverages masks to recover degraded features while enhancing cross-modal interactions, offering a valuable new perspective for image fusion task in challenging scenarios. The proposed task-coupled degradation-aware learning strategy exploits restoration priors to encourage the fusion network to learn cleaner representations, which is a well-motivated supervision scheme.

**Weaknesses:**

**Major**

(1) In Fig. 2, constraining the network directly with clean multimodal source images seems to improve generalization. However, intuitively, models trained on harder examples often generalize better to easier ones. Authors should clarify why the opposite appears beneficial here, and provide theoretical insights or empirical evidence.

(2) The mask is constructed based on Eq. (3). Please discuss whether this formulation captures the composition of more general fusion tasks, and detail its assumptions and potential failure modes.

(3)The evaluation on downstream tasks is not sufficiently comprehensive. Currently, only detection on M3FD is reported. Please add semantic segmentation comparisons on the MSRS dataset.

**Minor**

(1) Please standardize notation (case, subscripts/superscripts) throughout, including symbols in equations. Also, keep the term “Pseudo Ground Truth” consistent.

(2) The qualitative results in Fig. 6 are relatively small. please add higher resolution visual comparisons for adverse-weather cases in the appendix.

**Questions:**

(1) How is the Pseudo Ground Truth generator trained? Does it use off-the-shelf weights, or is it re-trained/fine-tuned on AWMM-100k? Please provide more training details and data splits.

(2) Can the proposed method generalize to other image fusion tasks, such as CT–MRI fusion in medical imaging?

---

> ### Author Response · Authors · 2025-11-20
> **Response to Weakness 1, 2 ,4 and 5 (About supervision approach, generality of the Mask formulation, standardized notations and higher-resolution visual comparisons)**
>
> *We sincerely thank Reviewer Gdro for the positive and thorough evaluation of our work. We have clarified the performance differences caused by various training strategies, supplemented experiments on downstream semantic segmentation tasks, and added more fusion examples, all of which significantly enhance the completeness and readability of the manuscript. We also appreciate the reviewer’s careful identification of formatting issues, which we have addressed point by point. Thank you very much for your time and effort.*
>
> **The revisions have been incorporated into the manuscript, and the latest version is now available for your review.**
>
> `W1. In Fig. 2, constraining the network directly with clean multimodal source images seems to improve generalization. However, intuitively, models trained on harder examples often generalize better to easier ones. Authors should clarify why the opposite appears beneficial here, and provide theoretical insights or empirical evidence.`
>
> We thank the reviewer for the question. In our Figure 2 example, there is no comparison between "easy" and "hard" cases. Instead, Figure 2 illustrates the model’s performance under two types of supervision: Pseudo Ground Truth and clean source images. The input source images are all degraded multi-modal data, so the task difficulty is identical in both cases. Supervising the model with clean source images enhances its generalization in standard scenarios, as the clean images encourage the algorithm to focus more on feature allocation and interaction rather than pixel-level restoration.
>
> `W2. The mask is constructed based on Eq. (3). Please discuss whether this formulation captures the composition of more general fusion tasks, and detail its assumptions and potential failure modes.`
>
> We would like to clarify that Equation 3 in the manuscript is presented to formalize the standard infrared-visible image fusion task. It can also represent a broader range of image fusion tasks, such as multi-focus image fusion. In multi-focus fusion, one technical branch is decision-map-based methods, which use a Mask to extract sharp regions from the foreground- and background-focused images and then combine them. For medical image fusion, the situation is similar to infrared-visible fusion, as the fusion results are also determined by the algorithm’s allocation of information from different modalities.
>
> Regarding failure cases, we have discussed them in Appendix.11. Specifically, under noisy conditions, the Mask generation strategy derived from Equation 3 can fail, which represents an issue we aim to address in future research.
>
> `W4. Please standardize notation (case, subscripts/superscripts) throughout, including symbols in equations. Also, keep the term “Pseudo Ground Truth” consistent.`
>
>
> Thank you for your valuable suggestion. We have carefully standardized the notation throughout the manuscript, including case, subscripts, and superscripts in all equations. Additionally, the term “Pseudo Ground Truth” is now used consistently across the paper.
>
>
> `W5. The qualitative results in Fig. 6 are relatively small. please add higher resolution visual comparisons for adverse-weather cases in the appendix.`
>
> Thank you for your comment. We have provided higher-resolution visual comparisons under adverse weather conditions, which can be found in Appendix Figures 13–15.

---

> ### Author Response · Authors · 2025-11-20
> **Response to Weakness 3, Question 1 and 2 (About semantic segmentation results,  Pseudo-GT generator trained, and generalization to other fusion tasks)**
>
> `W3. The evaluation on downstream tasks is not sufficiently comprehensive. Currently, only detection on M3FD is reported. Please add semantic segmentation comparisons on the MSRS dataset.`
>
> We thank the reviewer for the comment. We have added segmentation performance comparisons of all algorithms on the MSRS dataset. This information can be found in Appendix A.8, as detailed below.
>
>
> > ### Appendix A.8: Additional Downstream Task Evaluation: Semantic Segmentation
>
> Beyond evaluating different fusion methods on object detection to assess their ability to preserve salient target information, we also examine their capacity to retain global scene semantics through a semantic segmentation downstream task. Specifically, we employ BANet on the MSRS dataset to compare the segmentation accuracy of images generated by various fusion algorithms. As shown in the Table 8, our proposed method ranks within the top two across all six categories and achieves the highest overall mIoU. Although EMMA and Text-IF also show strong semantic retention capability, they struggle to maintain robustness under adverse weather degradations. In contrast, our method leverages a mask-guided learning strategy that effectively strengthens multi-modal feature interaction, enabling the extraction of more discriminative semantic information and substantially enhancing the semantic expressiveness of the fused results.
>
>
> > **Tab. Segmentation accuracy comparison on the MSRS dataset.**
> >
> > ---
> >
> > | Methods     | Pub.          | Background | Car   | Person | Bike  | Curve | Car Stop | Guardrail | Color cone | Bump  | mIoU  |
> > |------------|---------------|-----------|-------|--------|-------|-------|----------|-----------|------------|-------|-------|
> > | LRRNet     | *PAMI 23*     | 98.34     | 89.09 | 68.11  | 69.29 | 52.04 | 71.57    | 81.97     | 64.28      | 78.27 | 74.77 |
> > | Text-DiFuse| *NIPS 24*     | 98.45     | 88.96 | 71.04  | 71.94 | 60.05 | 72.23    | 84.03     | 65         | 78.59 | 76.7  |
> > | EMMA       | *CVPR 24*     | 98.53 | 90.36 | 74.65 | 71.61 | 64.61 | 74.49 | 83.73 | 65.61 | 76.36 | 77.77 |
> > | Text-IF    | *CVPR 24*     | 98.51     | 90.27 | 73.01 | 72.17 | 64.86 | 74.11 | 84.88 | 66.24 | 78.98 | 78.12 |
> > | GIFNet     | *CVPR 25*     | 98.4      | 88.98 | 71.13  | 71.92 | 59.19 | 74.11    | 78.05     | 65.3       | 64.62 | 74.63 |
> > | SAGE       | *CVPR 25*     | 98.38     | 88.8  | 69.53  | 70.37 | 57.13 | 71.21    | 86.06 | 65.08      | 76.9  | 75.94 |
> > | AWFusion   | *Arxiv 24*    | 98.52     | 89.5  | 71.8   | 71.75 | 64.17 | 74.18    | 83.67     | 66.01      | 77.53 | 77.46 |
> > | AMG-Fuse   | –             | **98.54** | 89.69 | 72.63  | **72.26** | 63.86 | 74.13 (2) | 85.89 (2) | **66.27** | **80.31** | **78.18** |
>
> `Q1. How is the Pseudo Ground Truth generator trained? Does it use off-the-shelf weights, or is it re-trained/fine-tuned on AWMM-100k? Please provide more training details and data splits.`
>
> We thank the reviewer for the comment. We have described the training settings for EMMA in Section Appendix.3. The details are as follows.
>
> > In summary, the goal of using “Pseudo Ground Truth’’ is to provide the network with a more reliable optimization trajectory during the early training stage. We adopt EMMA as the generator, not to seek the optimal “Pseudo Ground Truth’’, but to supply a reliable prior for facilitating convergence. Since EMMA has already been trained on existing multi-modal datasets, we fine-tune it on the clean subset of AWMM-100k comprising 1,000 images, while preserving its original weights. All training configurations strictly follow those reported in the original paper.
>
> `Q2. Can the proposed method generalize to other image fusion tasks, such as CT–MRI fusion in medical imaging?`
>
> Thank you for your comment regarding the generalization of our proposed method to medical image fusion tasks. Since CT–MRI fusion shares a similar nature with infrared–visible image fusion, our method can be effectively generalized to this domain. The table below compares our method with three state-of-the-art methods. As shown, our method achieves the best performance across all four evaluation metrics, demonstrating superior overall effectiveness.
>
>
> | Methods   | Pub.    | Q_MI     | Q_G      | Q_M      | VIF     | Q_CB     | SSIM    | Q^AB/F   |
> |-----------|-----------|---------|---------|---------|---------|---------|---------|---------|
> | EMMA      | CVPR 24   | 0.6718  | 0.5058  | 0.2470  | 0.2745  | **0.4929**  | **0.3029**  | 0.6489  |
> | GIFNet    | CVPR 25   | 0.6461  | **0.5717**  | 0.1325  | 0.1689  | 0.3134  | 0.1942  | 0.4195  |
> | SAGE      | CVPR 25   | 0.6597  | 0.5478  | 0.1450  | 0.2263  | 0.2637  | 0.2054  | 0.4244  |
> | AWG-Fuse  | –         | **0.7441**  | 0.4102  | **0.3725**  | **0.3120**  | 0.3597 (2)  | 0.2710 (2)  | **0.6509**  |

---

> ### Author Response · Authors · 2025-11-25
>
> We thank the reviewer for recognizing both our work and the rebuttal content. ***This affirmation clearly indicates that the earlier concerns have been fully resolved.***
>
> To further address your question, we conducted an ablation study by removing both MGLS and TDAS to evaluate the model trained solely with the fusion loss. The corresponding results are shown in the table below. As observed, training only with the fusion loss leads to a decrease across all metrics, with particularly notable declines in feature-related measures. This is because the model faces increased optimization difficulty and thus struggles to extract salient multi-modal features and restore fine details effectively.
>
> >---
> > | Methods           | Q_MI    | Q_G     | Q_M     | VIF    | Q_CB    | SSIM   | Q^AB/F  |
> > |------------------|--------|--------|--------|--------|--------|--------|--------|
> > | Only Fusion Loss | 0.3675 | 0.3673 | 0.5926 | 0.3340 | 0.4725 | 0.3835 | 0.4725 |
> > | AMG-Fuse         | **0.4084** | **0.4163** | **0.6940** | **0.3582** | **0.4879** | **0.4080** | **0.5184** |

---

### Official Review · Reviewer_cVus · 2025-10-31

**Soundness:** 3
**Presentation:** 3
**Contribution:** 3
**Rating:** 6
**Confidence:** 4

**Summary:**

The paper proposes a mask-guided multi-modality image fusion (MMIF) framework that jointly performs feature restoration and cross-modal interaction under adverse weather.
It introduces pseudo ground-truth to simplify training and a mask generation mechanism to measure each modality’s contribution during fusion.
A mask-guided cross-modal attention enables the network to focus on informative features, while degradation-aware learning strategies balance restoration and interaction.
Experiments on rain, haze, and snow datasets show that the proposed method achieves superior visual quality, quantitative performance, and downstream task results compared to state-of-the-art methods.

**Strengths:**

1. This work introduces a mask-guided framework (AMG-Fuse) that unifies feature restoration and modality interaction, addressing the limitations of two-stage “restoration + fusion” pipelines.

2. The Mask-Guided Feature Extraction Module (MFEM), Mask-Guided Learning Strategy (MGLS), and Task-Coupled Degradation-Aware Strategy (TDAS) are well-motivated and technically sound contributions that enhance cross-modal complementarity.

3. Extensive experiments on multiple adverse weather datasets (rain, haze, snow) and clean datasets demonstrate consistent improvements across quantitative and qualitative metrics.

4. The proposed method outperforms state-of-the-art methods in both degraded and ideal conditions, showing robustness and generalization ability.

5. The paper is well-organized, with detailed ablation studies, visualization analyses, and fair comparisons that support the claims effectively.

**Weaknesses:**

1. Although conceptually justified, the reliance on pseudo targets may still raise questions about the upper-bound constraint on model generalization and possible supervision bias not fully explored in the experiments. Though softened by decaying loss weights, the reliance on pseudo targets may introduce bias and constrain the model’s learning dynamics. Would this be addressed by theoretical or empirical analysis?

2. Could the authors provide a more extensive comparisons on FLOPs or runtime for efficiency analysis?

3. The method introduces Histogram Transformer and multi-head attention structures, yet the discussion of computational trade-offs versus the versions of lightweight fusion models remains underdeveloped.

4. The manuscript has neglected several related works, such as pseudo ground-truth in detection under adverse weather [1], image fusion under hazy condition [2], etc. More extensive literature review is recommended.

[1] Rethinking image restoration for object detection. 2022.
[2] Image dehazing by artificial multiple-exposure image fusion, 2018.

5. Some mathematical derivations (e.g., Equation 5 reformulation) and visualizations could be better contextualized for readers unfamiliar with modality-decoupled learning. The citation format of the paper should be revised.

The concerns are not vital and could be addressed before potentially accepted.

**Questions:**

Would the authors consider providing theoretical or empirical analysis to clarify whether the reliance on pseudo targets introduces supervision bias or constrains the model’s generalization capacity and learning dynamics?

---

> ### Author Response · Authors · 2025-11-20
> **Response to Weakness 1 and Question1 (About Pseudo-GT supervision bias)**
>
> *We sincerely thank Reviewer cVus for recognizing the novelty and performance of our work. ***Notably, Reviewer cVus also emphasized that “the concerns are not vital and could be addressed before potentially accepted,” which we greatly appreciate.*** Through point-by-point revisions and responses to the weaknesses you highlighted, the quality of our manuscript has been substantially improved. In particular, we have conducted a more in-depth analysis of the impact of Pseudo Ground Truth on model training, which has further strengthened our work. Below, we provide detailed point-by-point responses to your comments. Thank you very much for your time and effort.*
>
> **The revisions have been incorporated into the manuscript, and the latest version is now available for your review.**
>
> `W1. Although conceptually justified, the reliance on pseudo targets may still raise questions about the upper-bound constraint on model generalization and possible supervision bias not fully explored in the experiments. Though softened by decaying loss weights, the reliance on pseudo targets may introduce bias and constrain the model’s learning dynamics. Would this be addressed by theoretical or empirical analysis?`
>
> `Q1. Would the authors consider providing theoretical or empirical analysis to clarify whether the reliance on pseudo targets introduces supervision bias or constrains the model’s generalization capacity and learning dynamics?`
>
> We thank the reviewer for raising the important concern regarding the reliance on Pseudo Ground Truth. Specifically, whether Pseudo Ground Truth may limit the mode’s generalization upper bound or introduce potential supervisory bias during training. We have provided both theoretical and empirical analyses of these potential issues in Appendix A.3, summarized as follows:
>
> > ### Appendix A.3: The Discussion and Generation of Pseudo Ground Truth
>
> > While the above analysis suggests that Pseudo Ground Truth do not restrict the model’s capacity, we further validate this claim empirically. Specifically, we first used EMMA to generate fused results on clean multi-modal images and treated them as Pseudo Ground Truth. Then, we fed the same clean multi-modal inputs into our proposed model and directly compared the fusion outputs of both methods. According to the hypothesis, if our model were strictly limited by the modality distribution and interaction patterns provided by EMMA during training, then the performance of EMMA should represent the upper bound of our model. However, as shown in the Table 4, our method significantly surpasses EMMA across all three standard datasets. This evidence directly refutes the concern that Pseudo Ground Truth could restrict the upper-bound performance of model, demonstrating that our model is not constrained by the representational capacity of the Pseudo Ground Truth. Instead, it can further refine feature interactions and fusion strategies beyond the priors encoded in EMMA.
>
> > In summary, the goal of using Pseudo Ground Truth is to provide the network with a more reliable optimization trajectory during the early training stage. We adopt EMMA as the generator, not to seek the optimal Pseudo Ground Truth, but to supply a reliable prior for facilitating convergence. Since EMMA has already been trained on existing multimodal datasets, we fine-tune it on the clean subset of AWMM-100k while preserving its original weights. All training configurations strictly follow those reported in the original paper.
>
> > | Source | Methods | Pub. | Q_MI | Q_G | Q_M | VIF | Q_CB | SSIM | Q^AB/F |
> > |-----------|-------------|----------|------|------|-------|------|-------|-------|----------|
> > | MSRS | EMMA | *CVPR 24* | 0.6697 | 0.5624 | 0.7291 | 0.4165 | 0.5394 | 0.4746 | 0.6500 |
> > | MSRS | AMG-Fuse | – | **0.7594** | **0.5952** | **1.5051** | **0.4706** | **0.5809** | **0.5044** | **0.6749** |
> > | | | | | | | | | | |
> > | M3FD | EMMA | *CVPR 24* | 0.5646 | 0.4827 | 0.5595 | 0.3829 | 0.4880 | 0.4787 | 0.6032 |
> > | M3FD | AMG-Fuse | – | **0.7010** | **0.5376** | **1.2994** | **0.4713** | **0.4979** | **0.5069** | **0.6384** |
> > | | | | | | | | | | |
> > | LLVIP | EMMA | *CVPR 24* | 0.4437 | 0.5005 | 0.2314 | 0.3301 | **0.4559** | 0.4339 | 0.5969 |
> > | LLVIP | AMG-Fuse | – | **0.4684** | **0.6228** | **0.7938** | **0.3952** | 0.4479 | **0.4574** | **0.6907** |
>
> **The complete content can be found in Appendix A.3.**

---

> ### Author Response · Authors · 2025-11-20
> **Response to Weakness 2 and 3 (About lightweight variant)**
>
> `W2. Could the authors provide a more extensive comparisons on FLOPs or runtime for efficiency analysis?`
>
> `W3. The method introduces Histogram Transformer and multi-head attention structures, yet the discussion of computational trade-offs versus the versions of lightweight fusion models remains underdeveloped.`
>
> We are grateful to the reviewer for highlighting the computational resource requirements of our algorithm, which indeed warrant careful consideration in practical applications. While the histogram transformer module and depth-separable convolutions increase computational complexity and parameter count, our primary goal is to achieve high-quality multimodal fusion under complex and challenging conditions. Accordingly, the original model was not specifically designed for low-computational-power devices.
>
> Nonetheless, motivated by this concern, we explored a lightweight variant of our model, which is described in **Appendix A.9**.
>
> > ### Appendix A.9: Mask Learning on a Tiny Variant: Efficiency and Generalization
> >
> > In this study, we validate that the proposed mask learning strategy can significantly improve the learning efficiency of the network. It enables the model to better capture the dynamic interaction mechanisms between multimodal features while effectively suppressing overfitting risks.
> >
> > For lightweight models, whose parameter capacity is limited, the amount and diversity of perceptible features are substantially reduced compared with the original large-scale model. As a result, these models are more prone to learning static feature distributions, which may lead to overfitting or difficulties in model fitting. Under such circumstances, if the proposed mask learning strategy is truly effective, its advantages should become even more pronounced when applied to lightweight architectures.
> >
> > Motivated by this consideration, we introduce a lightweight structural redesign of the original model and construct a Tiny variant. Our test images have a resolution of 224×224, and the detailed descriptions are as follows:
> >
> > **Tiny Model Variant:**
> > - The initial channel dimension processed by the HTB is reduced from 48 to 16.
> > - The number of Transformer blocks is changed from [4, 6, 6, 8] to [1, 2, 2, 4].
> > - The number of attention heads is modified from [2, 4, 6, 8] to [1, 2, 2, 4].
> >
> > The resulting model contains 4.91M parameters with a total of 20.972 GFLOPs.
> >
> > We conducted a comprehensive evaluation of the lightweight model under three typical adverse weather scenarios, and the results are presented in Table 9. Since the main paper already provides a full quantitative comparison against all competing methods, the appendix reports only the average ranking of the lightweight AMG-Tiny model and all comparative methods across three adverse weather conditions.
> >
> > It can be observed that AMG-Tiny still achieves top-ranked performance under adverse weather conditions. Although its overall scores are slightly lower than those of Text-IF, it is worth noting that Text-IF has 249.707 GFLOPs and 89.014M parameters, whereas the proposed AMG-Tiny achieves reductions of approximately 11.9× in FLOPs and 18.1× in parameters, demonstrating a substantial efficiency advantage.
> >
> > **Efficiency and Inference Performance:**
> > - Original model: 242.03 GFLOPs, 59.74M parameters
> > - Tiny variant: 20.972 GFLOPs, 4.91M parameters
> > - Reductions: 11.54× (FLOPs) and 12.17× (parameters)
> >
> > During inference, the model processes a single $224\times224$ image in 40.302 ms on average (standard deviation 0.480 ms), corresponding to approximately 24.81 FPS.
> >
> >---
> > | Methods     | Pub.       | Restor. | QMI         | QG          | QM          | VIF         | QCB         | SSIM        | QAB/F       |
> > |------------|------------|---------|------------|------------|------------|------------|------------|------------|------------|
> > | LRRNet     | PAMI 23    | AdaIR   | 0.3606     | 0.2733     | 0.3810     | 0.2457     | 0.3767     | 0.1350     | 0.3668     |
> > | Text-DiFuse| NIPS 24    | AdaIR   | 0.3312     | 0.2875     | 0.3581     | 0.2329     | 0.4188     | 0.2606     | 0.3827     |
> > | EMMA       | CVPR 24    | AdaIR   | 0.4072     | 0.3844     | 0.5125     | 0.3065     | 0.4788     | 0.3697     | 0.4984     |
> > | Text-IF    | CVPR 24    | AdaIR   | 0.3476     | 0.4094     | 0.5578     | 0.3189     | 0.4800     | 0.3929     | 0.5150     |
> > | GIFNet     | CVPR 25    | AdaIR   | 0.2908     | 0.3127     | 0.3391     | 0.2741     | 0.4824     | 0.3553     | 0.3634     |
> > | SAGE       | CVPR 25    | AdaIR   | 0.3667     | 0.2950     | 0.4595     | 0.3382     | 0.4312     | 0.3175     | 0.4197     |
> > | AWFusion   | Arxiv 24   | x       | 0.3605     | 0.3723     | 0.5811     | 0.3152     | 0.4700     | 0.3721     | 0.4959     |
> > | AMG-Tiny   | –          | x       | 0.3713 (2) | 0.3851 (2) | 0.5422 (3) | 0.3327 (2) | 0.4776 (4) | 0.3887 (2) | 0.4965 (3) |
> >---
> **The complete content can be found in Appendix A.9.**

---

> ### Author Response · Authors · 2025-11-20
> **Response to Weakness 4 and 5 (About additional relevant literature and citation format)**
>
> `W4. The manuscript has neglected several related works, such as pseudo ground-truth in detection under adverse weather [1], image fusion under hazy condition [2], etc. More extensive literature review is recommended.`
>
> We thank the reviewer for this helpful suggestion. We have incorporated the two recommended references into the Introduction.
>
> `W5. Some mathematical derivations (e.g., Equation 5 reformulation) and visualizations could be better contextualized for readers unfamiliar with modality-decoupled learning. The citation format of the paper should be revised.`
>
> We thank the reviewer for the valuable comment. We have added contextual explanations in the algorithm section (Lines 209–232) to make the mathematical derivations and visualizations more accessible to readers unfamiliar with modality-decoupled learning, as detailed below.
>
> > In real-world scenarios, directly subtracting infrared images from visible images can lead to misleading or unstable behaviour. Under haze or snow, visible images often exhibit excessive or locally overexposed brightness, whereas infrared images show reduced contrast and lower pixel values. After network optimisation and degradation removal, the fused output enhances the structural and target information from the infrared component, resulting in a more stable ($Fuse - IR$) distribution that better reflects the true scene brightness. However, because the brightness bias in the denominator of Equation 4 is driven by degenerative factor form visible image rather than semantic content, degraded information dominates the weighting, violating the decoupling principle of mask and preventing it from capturing the actual modal distribution. Conversely, in night-time scenes, visible images contain minimal effective texture due to insufficient illumination, while infrared images remain relatively stable and may present stronger brightness cues. This causes ($VI - IR$) to be negative or near zero over large regions, making Equation 4 numerically unstable when generating the mask. To address this issue, we rewrite the expression of $M$ as follows:
>
> Furthermore, we have revised the reference format to ensure compliance with ICLR requirements.

---

### Author Response · Authors · 2025-12-04
**Summary of Author Feedback and Manuscript Revisions**

**Dear AC and SAC**

*Thank you for your time and effort. Below is a concise summary to help you quickly grasp our responses and the key contributions of our work.*

Firstly, we sincerely appreciate all the reviewers for recognizing the novelty and technical soundness of our work:

- **Reviewer cVus** acknowledged the proposed Mask-Guided Feature Extraction Module (MFEM), Mask-Guided Learning Strategy (MGLS), and Task-Coupled Degradation-Aware Learning Strategy (TDAS), describing them as *“well-motivated and technically sound contributions.”*
- **Reviewer Gdro** found our MGLS *effective and meaningful*, noting its ability to enhance cross-modal information interaction.
- **Reviewer AjCT** acknowledged our model *“overcomes the limitations of traditional methods relying on static modality distribution.”*

*In response to the reviewers’ concerns, we have provided detailed clarifications and conducted additional experiments. The major updates incorporated into the revised manuscript are as follows:*

---

**(1) Pseudo-GT limitation**

In response to Reviewer cVus’s concern that the Pseudo Ground Truth (Pseudo-GT) might constrain the model’s performance, we conducted additional quantitative comparisons with the pseudo-GT. The results show that our method outperforms the pseudo-GT on 20 out of 21 evaluation metrics across three datasets, demonstrating that the model is not bounded by pseudo-GT (***See Appendix A.3***).

**(2) Lightweight model**

In response to concerns from Reviewers cVus and AjCT about lightweight deployment, we conducted additional experiments on the this version. The results show that the lightweight model reduces FLOPs and parameter by 11.54× and 12.17×, respectively, while maintaining SOTA performance. This demonstrates that our model can be effectively deployed on low-power or resource-limited devices (***See Appendix A.9***).

**(3) Semantic segmentation performance**

To address Reviewer Gdro’s concern on downstream task performance, we conducted additional semantic segmentation experiments on the MSRS dataset over 9 categories. The results further validate the downstream benefits of our method (***See Appendix A.8***).

**(4) Real-world scenes evaluation**

Following Reviewer AjCT’s question about the model’s real-world scenes performance, we provide further clarification and experiments on the AWMM-100K real degradation dataset. Results show strong generalization to real degraded scenes (***See Table 2 in the main text***).

**(5) Comparison with more methods**

As suggested by Reviewer AjCT’s, we added two restoration methods (DA-CLIP and DehazeFormer) and two recent image fusion methods (UMCFuse and OmniFuse). (***See Appendix A.7 and our response to Reviewer AjCT’s Q3***).

---

All relevant changes in the revised manuscript are highlighted in blue.

---

**Final remarks**

1. Reviewer cVus noted that their concerns were not major and could be addressed before acceptance; after reading the response, they raised their rating (**score 6, raised to 8**) on Nov 26 before the “OpenReview Data Leak”.
2. Reviewer Gdro explicitly stated that their concerns were resolved.
3. Although Reviewer AjCT did not provide a follow-up response, we have addressed all their points comprehensively.

*We believe that our responses fully and substantively address all reviewer comments and have effectively improved the overall quality of the paper. We sincerely thank the reviewers and the AC for their careful feedback and valuable suggestions.*

**Best regards,**
**Authors 313**

---

### Meta-Review · Area_Chair_9P61 · 2026-01-07

**Summary:**

The paper proposes a mask-guided multi-modality image fusion framework that jointly performs feature restoration and cross-modal interaction under adverse weather conditions, leveraging pseudo ground truth and adaptive mask mechanisms to enhance both fusion quality and downstream task performance.

However, after reviewing the paper and the authors’ response, AC identifies significant shortcomings in the experimental design. Specifically, the evaluation protocol for adverse weather conditions raises concerns:

> “For the adverse weather experiments, we selected 1,000 images from each of the three distinct weather conditions (Snow, Rain, Haze) in the AWMM-100k dataset (Li et al., 2024b) for training, and 150 images for testing. In addition, we evaluated our method on 23 real degraded images from AWMM-100k to further validate its performance under authentic degradation.”

The test set—particularly the subset of only 23 real degraded images—is strikingly small, which substantially limits the statistical reliability and generalizability of the reported results. Moreover, the paper does not specify the criteria used for selecting these images, raising questions about potential bias or cherry-picking. Such experimental settings fall short of the rigor expected for a venue like ICLR.

**Reviewer Concerns:**

Most reviewers' concerns have been addressed in the rebuttal. However, the real-world generalization problem and lack of realism in the synthetic rain/snow degradation problem are not addressed due to insufficient testing data.

**Reviewer Scores:**

Reviewer cVus and Reviewer Gdro will be positive while Reviewer AjCT will be negative.

---

### Decision · Program_Chairs · 2026-01-26

Reject